# Effect of volcanic aerosol on stratospheric $NO_2$ and $N_2O_5$ from 2002-2014 as measured by Odin-OSIRIS and Envisat-MIPAS

Cristen Adams[1,2], Adam E. Bourassa[1], Chris A. McLinden[3], Chris E. Sioris[3], Thomas von Clarmann[4], Bernd Funke[5], Landon A. Rieger[1], and Douglas A. Degenstein[1]

[1]Institute of Space and Atmospheric Studies, University of Saskatchewan, Saskatoon, Canada.
[2]Alberta Environmental Monitoring and Science Division, Government of Alberta, Edmonton, Alberta, Canada
[3]Environment and Climate Change Canada, Downsview, Ontario, Canada.
[4]Karlsruhe Institute of Technology, Institute of Meteorology and Climate Research, Karlsruhe, Germany.
[5]Instituto de Astrofísica de Andalucía, CSIC, Granada, Spain

Correspondence to: Cristen Adams (cristenlfadams@gmail.com)

**Abstract.** Following the large volcanic eruptions of Pinatubo in 1991 and El Chichón in 1982, decreases in stratospheric $NO_2$ associated with enhanced aerosol were observed. The Optical Spectrograph and InfraRed Imaging Spectrometer (OSIRIS) likewise measured widespread enhancements of stratospheric aerosol following seven volcanic eruptions between 2002 and 2014, although the magnitudes of these eruptions were all much smaller than the Pinatubo and El Chichón eruptions. In order to isolate and quantify the relationship between volcanic aerosol and $NO_2$, $NO_2$ anomalies were calculated using measurements from OSIRIS and the Michelson Interferometer for Passive Atmospheric Sounding (MIPAS). In the tropics, variability due to the quasi-biennial oscillation was subtracted from the time series. OSIRIS profile measurements indicate that the strongest anticorrelations between $NO_2$ and volcanic aerosol extinction were for the 5-km layer starting ~3 km above the climatological mean tropopause at the given latitude. OSIRIS stratospheric $NO_2$ partial columns in this layer were found to be smaller than background $NO_2$ levels during these aerosol enhancements by up to ~60% with typical Pearson correlation coefficients of R ~ -0.7. MIPAS also observed decreases in $NO_2$ partial columns during periods affected by volcanic aerosol, with percent differences of up to ~25% relative to background levels. An even stronger anticorrelation was observed between OSIRIS aerosol optical depth and MIPAS $N_2O_5$ partial columns, with R ~ -0.9, although no link with MIPAS $HNO_3$ was observed. The variation of OSIRIS $NO_2$ with increasing aerosol was found to be consistent with simulations from a photochemical box model, within estimated model uncertainty.

## 1 Introduction

Major volcanic eruptions can increase levels of sulfate aerosols in the stratosphere, which provide surfaces on which heterogeneous chemical reactions take place. This, in turn, can affect photochemistry with some of the largest impacts expected for the partitioning of reactive nitrogen, $NO_y$ species (where $[NO_y]=$ $[NO]+[NO_2]+[HNO_3]+2[N_2O_5]+[ClONO_2]+[BrONO_2]$, e.g., Coffey, 1996). The two key heterogeneous reactions that compete with gas-phase chemistry at all stratospheric temperatures are (1) and (2) below (e.g., Cohen and Murphy, 2003). In the presence of volcanic aerosol, the rate of $N_2O_5$ conversion to $HNO_3$ increases as shown in Eq. (1):

$$N_2O_5 + H_2O \xrightarrow{Aerosol} 2HNO_3 \quad , \qquad (1)$$

leading to an increase in levels of $HNO_3$, and a decrease in levels of $N_2O_5$. Since $N_2O_5$ is a reservoir species for $NO_x$ (where $[NO_x] = [NO_2] + [NO]$), levels of $NO_x$ also decrease. The hydrolysis of $BrONO_2$, shown in Eq. (2):

$$BrONO_2 + H_2O \xrightarrow{Aerosol} HOBr + HNO_3 \quad , \qquad (2)$$

can also lead to decreased levels of $NO_2$ in the lower stratosphere, which are particularly significant toward high latitudes in the summer (Randeniya et al., 1997). The hydrolysis of chlorine nitrate can also play a significant role inside the polar vortex, (Wegner et al., 2012), but is not considered here because data inside the polar vortex were not analyzed.

Following the large 1982 and 1991 El Chichón and Pinatubo volcanic eruptions, several studies measured significant decreases in $NO_2$. Total columns of $NO_2$ measured by ground-based instruments decreased by ~15-70% following these eruptions (Coffey, 1996; Johnston et al., 1992; Koike et al., 1993; Mills et al., 1993). Fahey et al. (1993) measured in situ $NO_x/NO_y$ aboard aircraft following Pinatubo and found that $NO_x/NO_y$ decreased with increasing aerosol surface area, with a saturation effect toward larger aerosol surface areas.

The effects of the Pinatubo eruption on $HNO_3$ and $N_2O_5$ were also assessed in several studies. Some studies noted increases in $HNO_3$ and attributed these increases to Eq. (1), but this was not consistently observed across various ground-based, in situ, and satellite datasets (e.g., Coffey, 1996; Rinsland et al., 1994). Rinsland et al. (1994) found decreases in $N_2O_5$ following Pinatubo, which is also implied by Eq. (1).

Berthet et al. (2017) assessed the impact of the more recent Sarychev eruption in June 2009 on lower stratosphere chemistry using remote sensing, in situ measurements, and model calculations. Measured profiles of $NO_2$ and aerosol extinction were anti-correlated in the lower stratosphere following the eruption, with layers of enhanced aerosol coinciding with smaller $NO_2$

mixing ratios. Using model calculations, they estimated that $NO_2$ decreased by ~45% and $HNO_3$ increased by ~11% over the August-September 2009 period below 19 km following the volcanic eruption.

The Optical Spectrograph and InfraRed Imaging System (OSIRIS) has observed enhancements in stratospheric aerosol from multiple volcanoes since it began taking measurements in 2001 (Bourassa et al., 2012a). The effect of these more recent
volcanoes on stratospheric $NO_2$, also measured by OSIRIS, is investigated in the present study. Stratospheric $NO_2$, $HNO_3$, and $N_2O_5$ from the Michelson Interferometer for Passive Atmospheric Sounding (MIPAS) are also considered. Variations in $NO_2$, $HNO_3$ and $N_2O_5$ with aerosol are also studied using a photochemical box model. This coarse resolution study cannot be used to understand short-term processes in the days immediately following a volcanic eruption. Instead, it can be used to understand the longer-term effect of volcanic aerosol in the months following an eruption.

This paper is organized as follows. The OSIRIS, MIPAS, and photochemical model datasets are described in Sect. 2. Monthly average $NO_2$, $HNO_3$, and $N_2O_5$ anomalies and background values are calculated from these data using the methodology given in Sect. 3. The relationships between these anomalies and volcanic aerosol measured by OSIRIS are presented in Sect. 4, with conclusions given in Sect. 5.

## 2. Satellite and model datasets

**2.1 OSIRIS MART aerosol extinction and $NO_2$**

OSIRIS (Llewellyn et al., 2004; McLinden et al., 2012) is a Canadian satellite instrument on-board the Odin spacecraft (Murtagh et al., 2002), which was launched on 20 February 2001 into a sun-synchronous orbit at ~600 km altitude and a descending node equatorial crossing time of ~06:30 local time (LT). OSIRIS measures limb-scattered radiances from 82°S to 82°N, with nearly full coverage in the summer hemisphere.

The OSIRIS Multiplicative Algebraic Reconstruction Technique (MART) v5.07 $NO_2$ (Bourassa et al., 2011) and aerosol extinction at 750 nm (Bourassa et al., 2007) data products were used for this study. Data were collected by the optical spectrograph, which measures from 280-810 nm, with a ~1 nm spectral resolution using an optical grating and a charge-coupled device detector. The SASKTRAN spherical forward model is used in the inversion and accounts for multiple scattering and ground albedo (Bourassa et al., 2008). OSIRIS MART aerosol extinction is consistent with SAGE III to ~10%
(Bourassa et al., 2012b) and to within ~20% with SAGE II, although the conversion to 525 nm adds uncertainty (Rieger et al., 2015). The OSIRIS MART $NO_2$ data product is consistent with the OSIRIS Chalmers $NO_2$ data product (Bourassa et al., 2011).

For both aerosol extinction and NO$_2$, OSIRIS data from the descending portion of the orbit, with solar zenith angle (SZA) less than 88° were used in this analysis. For aerosol, an extinction threshold greater than $2\times10^{-3}$ km$^{-1}$ was used to terminate the profiles at lower altitudes. This excludes some lower stratospheric altitudes where an aerosol saturation effect occurs in fresh volcanic plumes (Fromm et al., 2014). Similarly, values of NO$_2$ greater than $5\times10^9$ mol/cm$^3$ were removed from the profiles.

5    Data below the thermal tropopause, calculated using lapse rates from the National Center for Environmental Prediction (NCEP) reanalysis data (Kalnay et al., 1996), were excluded. As discussed further in Sect. 2.3, below, in order to account for the diurnal variation of NO$_2$, a photochemical model (Brohede et al., 2008; McLinden et al., 2000) was used to scale all NO$_2$ profiles to a common local time of 06:30 LT. Profiles for SZA greater than 88° at 06:30 LT were also excluded from the analysis in order to prevent scaling data to periods near sunrise when NO$_2$ is varying rapidly.

10  **2.2 MIPAS IMK/IAA NO$_2$, N$_2$O$_5$, and HNO$_3$**

MIPAS (Fischer et al., 2008) is on board the Environmental Satellite (Envisat), which was launched on 1 March 2002 into a Sun-synchronous polar orbit at 800 km altitude with 14.4 orbits per day. MIPAS measured limb radiances in the mid-infrared from 4.1-14.7 µm (685-2410 cm$^{-1}$) until communication with the satellite was lost in April 2012.

For this study, N$_2$O$_5$, HNO$_3$, and NO$_2$ from retrievals performed by the Institute of Meteorology and Climate Research (IMK) 15  and Instituto de Astrofisica de Andalucia (IAA) were used. The version V5R_NO2_220/V5R_NO2_221, V5R_N2O5_220/V5R_N2O5_221, V5R_HNO3_224/V5R_HNO3_225 data from January 2005 to April 2012 were considered. Prior to 2005, data are available, but MIPAS operated with a different spectral resolution and only minor volcanic eruptions occurred. Retrievals for NO$_2$ (Funke et al., 2005, 2014), N$_2$O$_5$ (Mengistu Tsidu et al., 2004), and HNO$_3$ are performed using a constrained multiparameter nonlinear least squares fitting of measured spectra with modelled ones (von 20  Clarmann et al., 2009). Data unaffected by clouds and with diagonal terms of the averaging kernel greater than 0.03 were used for the analysis. Only daytime measurements (SZA less than 88°), taken at 10:00 LT, were used for consistency with OSIRIS.

**2.3 Photochemical modelling**

A stratospheric photochemical box model (Brohede et al., 2008; McLinden et al., 2000) was used to help interpret the satellite data. The model is constrained with climatological profiles of ozone and temperature. Long-lived species (N$_2$O, CH$_4$, H$_2$O) 25  and families (NO$_y$, Cl$_y$, Br$_y$) are based on a combination of three-dimensional model output or tracer-correlations. All remaining species are calculated to be in a 24-hour steady-state by integrating the model for as many as 30 days, but where the model remains fixed on the original specified Julian day. Heterogeneous chemistry on background stratospheric sulfate aerosols is included, but polar stratospheric clouds are not included. Brohede et al. (2008) demonstrated that this model can accurately simulate stratospheric nitrogen partitioning.

The model is typically used for two purposes: (i) to adjust the local time of the OSIRIS measurements to a common value through a photochemical scaling factor (e.g., Brohede et al., 2008, 2007), and (ii) to model the behavior of $NO_2$ and other species for varying levels of aerosol. In this latter application, the aerosol surface area, $SA$ ($\mu m^2/cm^3$), is adjusted so that it matches the extinction coefficient, $k$ ($km^{-1}$), measured by the OSIRIS instrument, using the expression,

$$SA(z) = \frac{4 \cdot 1000}{\bar{Q}} \cdot k(z) \quad , \qquad (3)$$

derived in Appendix A, where $\bar{Q}$ is effective scattering efficiency and the factor of 1000 is required to adjust between the common units of extinction ($km^{-1}$) and surface area density ($\mu m^2/cm^3$). A scattering efficiency of 0.40 was calculated using Mie theory for background spherical sulfate particles (based on a log-normal distribution with size parameters of $r_g$=0.08 µm and $\sigma_g$=1.6). However, volcanic eruptions alter the size distribution, as $SO_2$ rapidly forms sulfuric acid, which can condense to form new small particles or increase the size of existing ones. This change in size distribution will affect the scattering efficiency, but the sign of this change is unknown. For example, two months after the Kasatochi eruption, there was a shift in the ambient size distribution toward smaller particles (Sioris et al., 2010) whereas Sarychev led to a shift toward larger particles (O'Neill et al., 2012). As a result of these mixed findings, we elected to keep the scattering efficiency constant, using $SA = 10000 \cdot k$, but tested a non-linear $SA$ dependence on $k$ and included it in the model uncertainty estimates, as described in the paragraph below.

Table 1 summarizes the parameters used to estimate uncertainty in the modelled $NO_2$ percent differences. The model calculations were repeated, successively varying one key geophysical input or assumption, in order to assess its impact on the results. In total, eight such sensitivity calculations were performed and their individual $NO_2$ percent difference values were added in quadrature to estimate a total sensitivity or uncertainty. To account for potential errors and variability over 2002-2015 in our background $SA$, we scaled 10000 by factors of 3 and 1/3. The large factor is based on the sensitivity of scattering efficiency to the aerosol size parameters for the particle sizes and wavelengths considered here. For example, a change in effective radius by a factor 2 leads to a change in scattering efficiency by a factor of 3 (see Hansen and Travis (1974) Fig. 8). The dependence $SA$ on $k$ was estimated by $SA \propto k^p$ with $p$=1.3 for an increase in particle size following an eruption, and $p$=0.7 for a decrease, analogous to that from Thomason et al. (1997). In order to account for uncertainty and variability in the climatological input profiles, ozone, $NO_y$, and temperature profiles were perturbed by +10%, +20%, and +5 K, respectively. Surface albedo was changed from the original model setting of 0.1 to 0.3. For ozone, $NO_y$, temperature, and albedo, half the perturbation was applied as a systematic error, affecting both the monthly $NO_2$ and background $NO_2$, and half the perturbation was applied as a random error, which would account for year-to-year variability between conditions for background and monthly $NO_2$. The $NO+O_3$ reaction rate and $NO_2$ absorption coefficient were both perturbed by 10%, and applied as a systematic error only.

# 3 Calculation of monthly averages, anomalies, and background levels

## 3.1 OSIRIS and MIPAS

Volcanic eruptions and periods affected by volcanic aerosol were identified using monthly averages of latitude-binned OSIRIS aerosol extinction. In order to isolate the effect of volcanic aerosol on $NO_2$, $HNO_3$, and $N_2O_5$, anomalies and background levels in the absence of volcanic aerosol were estimated for each month and latitude. Background levels are the monthly mean values in the time series at a given latitude for months that were not affected by volcanic aerosol, based on a threshold for OSIRIS aerosol measurements. These calculations are described in further detail below.

Partial vertical column densities (VCDs) and partial aerosol optical depth (AOD), as well as vertical profiles were considered. Partial columns were used, instead of, e.g., volume mixing ratios at a fixed altitude, because the largest observed aerosol extinction ratios related to volcanic aerosol were observed at different altitude layers for different latitudes and times. The partial column altitude range was selected to include most of these large extinction ratios. Furthermore, MIPAS $NO_2$ measurements have low resolution at the altitudes affected by volcanic aerosol and therefore are better presented as partial columns. Calculations were made for each month and latitude, and for the profiles, each altitude layer was considered separately. Latitudes south of 50°S were excluded from the analysis because there is no evidence of volcanic aerosol at these southern hemisphere high latitudes and OSIRIS AODs are dominated by seasonal variation. The steps for these calculations are described in the paragraphs below.

Monthly average profiles of aerosol extinction, and of $NO_2$, $N_2O_5$, and $HNO_3$ number densities were calculated in 10° latitude bins. At least five measurements were required for each bin. Partial column AOD and partial VCDs of $NO_2$, $N_2O_5$, and $HNO_3$ were all calculated from the sum of these monthly mean profiles for a 5-km altitude range starting at ~3 km above the climatological mean NCEP thermal tropopause at each latitude. If a profile did not have valid data over all five measurement layers, it was not included in the analysis. This altitude range typically corresponded to the highest levels of volcanic aerosol observed by OSIRIS. Bins were tested for smaller latitude and time ranges, but yielded similar ranges of AODs, suggesting that smaller bin sizes did not capture more detailed processes in the volcanic plume.

MIPAS volume mixing ratio profiles were converted to number densities using MIPAS temperature and pressure profiles. Note the calculated VCDs for OSIRIS and MIPAS are offset by 0.5 km because their measurement altitude grids are offset by 0.5 km. MIPAS degrees of freedom for signal (DOFS) were calculated from the trace of the averaging kernel over the partial column altitude range. MIPAS VCDs with DOFS less than 0.5 were excluded from the analysis, leaving mean DOFS of 0.7 for $NO_2$, 1.8 for $N_2O_5$, and 2.1 for $HNO_3$.

Bins affected by volcanic aerosol were identified using thresholds based on OSIRIS aerosol extinction measurements. For the partial AODs, this threshold was set at $2\times10^{-3}$, which was approximately the 75[th] percentile of monthly mean partial column AODs across all latitudes for 50°S to 80°N. For the profiles, the threshold was set using an extinction ratio = 1.2, which was approximately the 90-95[th] percentile of monthly mean extinction ratios across all latitudes for 50°S to 80°N. The extinction ratio (OSIRIS-measured extinction divided by the Rayleigh extinction) has less dependence on altitude than the extinction and was calculated using air density profiles from European Centre for Medium-Range Weather Forecasts analysis data.

In order to remove the seasonal variation from the $NO_2$ time series, the $NO_2$ anomaly ($dNO_2$) was calculated for each bin of the monthly mean $NO_2$ VCDs as follows,

$$dNO_2(y, m, lat) \ = \ NO_2(y, m, lat) - \overline{NO_2^{No\,Volc}(m, lat)} \qquad , \qquad (4)$$

where $NO_2(y, m, lat)$ is the $NO_2$ VCD for the given year/month/latitude bin, and $\overline{NO_2^{No\,Volc}(m, lat)}$ is the mean $NO_2$ VCD for the given month/latitude across all years for bins that were not affected by volcanic aerosol, as determined by the AOD threshold. For the profile analysis, the $NO_2$ anomaly was calculated separately at each altitude from $NO_2$ number densities, with bins affected by volcanic aerosol identified using the extinction ratio threshold.

The quasi-biennial oscillation (QBO), with a mean period of ~28 months, is the dominant internal mode of climatic variability in the tropical stratosphere (see review by Baldwin et al., 2001). $NO_2$ can vary by more than 25% in the tropics near the tropopause due to QBO (Hauchecorne et al., 2010), which is on the same order of magnitude as the variation of $NO_2$ observed during periods of enhanced volcanic aerosol in this study. Therefore, the QBO was fit using a robust regression to the $NO_2$ anomaly time series for each latitude bin between 40°S to 40°N, using an approach similar to Randel and Wu (1996). Bins that were affected by volcanic aerosol were excluded from the fit. The fit was only performed if at least ten $NO_2$ anomaly values were available, and included the first two principal components of QBO, calculated with stratospheric winds from http://www.geo.fu-berlin.de/en/met/ag/strat/produkte/qbo/index.html (Naujokat, 1986). The two QBO principal components and 40°S to 40°N latitude range were selected based on the results of Bourassa et al. (2014). The QBO fits were subtracted from the $NO_2$ anomaly time series for each latitude bin between 40°S to 40°N. For the profile analysis, this procedure was applied separately at each altitude.

An example of the QBO fitting procedure is shown for 20.5 km at 0° latitude in Figure 1. The time series of extinction ratios shows several distinct periods with extinction ratios above the extinction ratio threshold of 1.2 (panel a). The QBO fit to the $NO_2$ anomaly time series includes only data collected during time-periods with extinction ratio less than 1.2 (panel b) in order to avoid fitting out some of the variability due to volcanic aerosol. The fit includes a constant and the two QBO principal

component terms only. After subtracting the fit, the NO$_2$ anomaly time series has stronger negative anomalies during the periods with enhanced volcanic eruptions (panel c).

Background NO$_2$ profiles and VCDs were estimated for periods that were not affected by volcanic aerosol, as identified using the OSIRIS aerosol extinction ratio and partial column AOD thresholds defined above. For mid-latitudes and high latitudes (50°S and 50°N – 80°N), $\overline{NO_2^{No\,Volc}(m, lat)}$ was used directly for background NO$_2$. For 40°S – 40°N, the fitted response to the QBO, as illustrated by the red dashed line in Figure 1b, was added to $\overline{NO_2^{No\,Volc}(m, lat)}$ in order to estimate the variation of background NO$_2$ with the QBO. The addition of the QBO signal had a minor impact on this analysis (~5-10%). For example, the background NO$_2$ VCD for January at 0° is the average of all January VCD measurements at 0°, except for the measurements taken in 2003, 2010, and 2012, when volcanic aerosol was enhanced. The QBO signal was added to the background NO$_2$ to account for year-to-year variations in background NO$_2$ (e.g., January 2008 background is slightly different than January 2009 due to QBO).

NO$_2$ was presented as a percent difference ($100\% \times \frac{NO_2\,Anomaly}{Background\,NO_2}$) for all figures and calculation of correlation coefficients in this study. Replacing percent difference NO$_2$ with the NO$_2$ anomaly has a minor influence on the shape of scatter plots and Pearson correlation coefficients (R) between NO$_2$ and aerosol extinction.

For N$_2$O$_5$ and HNO$_3$, anomalies and background values were calculated using the same approach as for NO$_2$. For aerosol extinction profiles and partial AODs, variations due to seasonal cycles are small and variations due to QBO in the tropics are less than 10% for 20-26 km (Hommel et al., 2015), while volcanic perturbations in aerosol extinction are often greater than 100%. Therefore, monthly averages of aerosol extinction and partial AODs were used directly in this analysis, without calculation of relative anomalies.

## 3.2 Photochemical model

A similar approach was used to assess variations in modelled aerosol extinction, NO$_2$, N$_2$O$_5$, and HNO$_3$ partial VCDs and profiles. The photochemical model was run monthly at 10° latitude intervals for a range of aerosol surface areas at the approximate OSIRIS measurement time (06:30 LT). For the column amounts, partial VCDs and partial column AODs were calculated over the 5 km layer, starting ~3 km above the tropopause for each of the model runs. Then, for each latitude and month, percent differences in NO$_2$, N$_2$O$_5$, and HNO$_3$ partial VCDs were calculated for a range of partial column AODs using ($100\% \times \frac{NO_2\,(AOD)-Background\,NO_2}{Background\,NO_2}$), where $NO_2(AOD)$ is the modelled NO$_2$ for the AOD/month/latitude and $Background\,NO_2$ is the modelled NO$_2$ for the given month/latitude, interpolated to the background partial column AOD. The background partial column AOD was calculated from OSIRIS measurements at the given latitude, for AODs that were less than $2\times10^{-3}$. For the

profiles, a similar interpolation procedure was used at each altitude layer using OSIRIS-measured aerosol extinction and the extinction ratio threshold of 1.2 to identify influence from volcanoes.

### 3.3 Conversions between partial column AODs, aerosol extinction, and extinction at various wavelengths

All AODs and aerosol extinctions presented here are for 750 nm, which is the wavelength of the OSIRIS retrievals. The partial column AODs are for a 5 km altitude range, and therefore can be related to the mean extinction ($km^{-1}$) over the given altitude range by dividing the partial column AOD by 5. In order to convert aerosol extinctions from 750 nm to other typical wavelengths, the conversion factors given in Table 2 can be used.

## 4 Results

### 4.1 $NO_2$, $N_2O_5$, and $HNO_3$ VCDs

We first examine an example of the modelled variation of $NO_2$, $N_2O_5$, and $HNO_3$ partial VCDs with volcanic aerosol, shown in Figure 2 for 60°N in August. This latitude was affected by enhanced volcanic aerosol in August from the 2009 Sarychev Peak and the 2011 Nabro eruptions, as discussed in further detail below. The maximum partial column AOD observed by OSIRIS at this latitude, $8x10^{-3}$, is indicated on the figure. Levels of $NO_2$ decrease strongly in the presence of larger partial column AODs, reaching percent differences of -45% relative to background levels for AOD=$8x10^{-3}$ at the approximate local time of OSIRIS measurements. At the MIPAS local time, reductions in $NO_2$ are slightly smaller, reaching -40% for AOD=$8x10^{-3}$. Percent differences in $N_2O_5$ decrease even more steeply than $NO_2$, reaching up to -86% relative to background levels for AOD=$8x10^{-3}$. $HNO_3$ increases slightly with partial column AOD, but is only +5% higher than background levels for AOD=$8x10^{-3}$. This is because $HNO_3$ is the dominant $NO_y$ species in the lower stratosphere. Therefore, even major changes in the partitioning due to heterogeneous chemistry on sulfate aerosol would have only a marginal relative impact on $HNO_3$.

Turning to the measurements, seven periods with volcanic aerosol enhancements were identified in the OSIRIS AOD time series, most of which were associated with negative $NO_2$ anomalies. This is apparent in the time series of AOD and percent difference $NO_2$ VCDs, shown in Figure 3 and summarized in Table 3. Note that only volcanoes with clear signals in the OSIRIS AODs are identified here, and therefore this is not a comprehensive list of all volcanoes known to have influenced the stratosphere. Höpfner et al. (2015) provide a list of volcanoes identified in MIPAS $SO_2$.

The OSIRIS $NO_2$ percent differences compared to background levels are largely negative during the AOD enhancements (thick black contours) and return to background levels when AODs decrease. This is consistent with reduction in $NO_2$ due to heterogeneous chemistry on the surface of sulfate aerosol. In the MIPAS data, $NO_2$ anomalies tend to be negative during

periods affected by volcanic aerosol. However, this relationship is weaker than observed with the OSIRIS $NO_2$ data and MIPAS measurements are not available for the largest observed OSIRIS AODs, as they did not meet the filtering criteria for this study. The variability of $NO_2$ outside periods of volcanic aerosol (standard deviation of the $NO_2$ anomaly divided by the mean background levels of $NO_2$ in each latitude bin) was less than 14% for both OSIRIS and MIPAS.

5    Figure 4 shows the correlation between $NO_2$ VCD percent differences and AOD for all times/latitudes. For OSIRIS, there appears to be a negative linear relationship, with R=-0.68. For larger AODs (greater than $4x10^{-3}$), OSIRIS VCDs are ~20-60% lower than under background conditions. For MIPAS, the relationship between $NO_2$ percent difference and AOD is weaker, with R = -0.37. When only MIPAS data from 40°N to 80°N are considered, the anticorrelation is somewhat stronger, with R = -0.50. Compared with OSIRIS, there are fewer monthly average MIPAS measurements available when AODs are high, which likely contributes to the weaker correlation. For AODs greater than $4x10^{-3}$, 18 out of 19 latitude-binned MIPAS VCDs have negative percent differences, typically in the range of ~5-25%.

The effect of volcanic aerosol on $NO_2$ is smaller for MIPAS than for OSIRIS. From the model simulations in Figure 2 it appears that the difference in local time of the measurements, 10:00 LT vs. 06:30 LT, can explain only a small part of this difference. These discrepancies could not be attributed to differences in sampling between OSIRIS and MIPAS, since MIPAS and OSIRIS both sample throughout the monthly 10° latitude bins. MIPAS measurements are not clustered in parts of the bin where smaller OSIRIS AODs were observed. A larger contributor to the smaller MIPAS $NO_2$ anomalies is a damping effect as many of the VCDs had DOFS less than 1, with smaller DOFS for larger OSIRIS partial column AODs. For OSIRIS partial column AODs greater than $5x10^{-3}$, the average MIPAS DOFS was ~0.6. The sub-optimal DOFS is accompanied by coarse altitude resolution which smooths $NO_2$ from higher altitudes where aerosol levels are not enhanced. In other words, the coarse altitude resolution leads to a smaller amplitude in the local $NO_2$ variation in the MIPAS data. In order to test this, representative MIPAS averaging kernels were applied to the OSIRIS $NO_2$ percent difference profiles at 50°N and 0° latitudes. Representative averaging kernels were used because MIPAS $NO_2$ is retrieved in the logarithmic domain and the averaging kernel thus refers to the logarithm of the mixing ratio. By applying the averaging kernel directly to the percent difference profile, the MIPAS a priori profile does not need to be included in the calculations. The magnitude of the largest percent differences in the $NO_2$ percent difference profiles decreased from approximately -45% in the original OSIRIS profiles to approximately -30% in the smoothed OSIRIS profiles, demonstrating this damping effect. These tests did not account for variation of MIPAS DOFS with partial column AOD. The smaller MIPAS DOFS observed for larger OSIRIS partial column AOD would lead to further damping of the MIPAS $NO_2$ percent differences. Also, correlation due to an $NO_2$ retrieval dependency on aerosol is expected to be small for OSIRIS because of the differential nature of the measurement and the spectral proximity of the selected strongly and weakly absorbing wavelengths (Bourassa et al., 2011).

The relationship between the MIPAS $N_2O_5$ anomaly and OSIRIS aerosol is also shown in Figure 3 and Figure 4. Strong anticorrelation between the $N_2O_5$ anomaly and AOD is observed, with R = -0.86 for 50°S to 80°N and R = -0.90 for 40°N to 80°N. The percentage decrease in $N_2O_5$ cannot be inferred due to the known low bias in the MIPAS data at these altitudes. Despite this low bias, the DOFS of ~1.8 suggest that real variability in $N_2O_5$ is observed.

MIPAS $HNO_3$ VCDs were also considered in this analysis (not shown here) using the same methodology as for $NO_2$ and $N_2O_5$, but no relationship with partial column AOD was apparent in the time series, with |R| less than 0.2 for most latitudes and altitude ranges considered. This is consistent with results from the photochemical model which suggest that $HNO_3$ should increase by less than 10% relative to background levels for all latitudes and partial column AODs observed in this study. Such small relative increases in $HNO_3$ would be difficult to observe over background variability.

Scatter plots for OSIRIS AOD versus $NO_2$ VCD percent difference, at latitudes where at least one AOD greater than $3\times10^{-3}$ was recorded, are shown in Figure 5. At all latitudes, higher AODs are associated with lower levels of $NO_2$, with R ranging from -0.40 to -0.85 at the various latitudes. The larger AODs are all measured following volcanic eruptions (Table 3). The modelled $NO_2$ percent differences, interpolated to the OSIRIS month, latitude, and partial column AOD are also shown. The modelled data agree well with the OSIRIS measurements and are within the estimated model errors for most OSIRIS data
points.

Other species are expected to be affected by increased aerosol, including $BrONO_2$, from Eq. (2), and by extension, BrO, where BrO is also measured by OSIRIS (McLinden et al., 2010). An analysis of the OSIRIS BrO product indicated no significant impact following the eruptions. This is consistent with a model-estimated increase in BrO of only 5-10% for AOD=$8\times10^{-3}$ (not shown here), and the reduced sensitivity of the OSIRIS BrO product below 20 km.

**4.2 OSIRIS $NO_2$ profiles**

The OSIRIS $NO_2$ and aerosol extinction profiles were used to assess the altitude range over which levels of $NO_2$ decreased in the presence of volcanic aerosol. The MIPAS $NO_2$ and $N_2O_5$ profiles were not considered for this purpose because of limitations in the vertical resolution in the lower stratosphere.

The time series of OSIRIS extinction ratio and OSIRIS and modelled $NO_2$ percent difference profiles are shown for 0° in
Figure 6. Negative $NO_2$ anomalies are apparent during the periods of enhanced aerosol for altitudes between ~16-24 km, with maximum decreases in $NO_2$ typically at ~20 km in both the OSIRIS and model datasets. These altitudes coincide approximately with the largest observed extinction ratios. OSIRIS measured percent differences in $NO_2$ of up to ~ -50% after the combined 2006 Soufrière Hills and Rabaul volcanoes, up to ~ -40% after the 2014 Kelut volcano, up to ~ -35% after the 2011 Nabro volcano, and up to ~ -25% after the 2005 Manam volcano. In the model dataset, similar qualitative features are

observed, but with somewhat smaller reductions in $NO_2$. Modelled percent differences reach up to ~ -20% after the Nabro and Manam volcanoes and up to ~ -30% after the Soufrière Hills/Rabaul and Kelut volcanoes. This is consistent with the modelling comparisons of VCDs and partial column AODs (Figure 5), in which modelled $NO_2$ percent differences at 0° latitude are biased somewhat low compared with OSIRIS. Only the aerosol extinction is varied inter-annually in the model, while other parameters such as $NO_y$, ozone, and temperature are from monthly climatologies, which is likely why the model displays less variability than OSIRIS during non-volcanic periods.

At 50°N, both OSIRIS and modelled negative $NO_2$ anomalies are similarly related to the times and altitudes of enhanced levels of aerosol, as shown in Figure 7. Decreases in $NO_2$ of up to ~ -50% are observed after the 2009 Sarychev Peak volcano, up to ~ -40% after the 2011 Nabro volcano, and up to ~ -30% after the 2008 Kasatochi/Okmok volcanoes. The modelled percent difference profiles are similar to the OSIRIS data, reaching ~ -40% for the Sarychev Peak and Nabro volcanoes and ~ -30% for the Kasatochi/Okmok volcanoes. The observed and modelled decreases in $NO_2$ for the Sarychev Peak eruption are consistent with Berthet et al. (2017), who calculated ~45% decrease below 19 km using model calculations.

The OSIRIS-measured variability in $NO_2$ outside of the periods of volcanic aerosol (standard deviation of the $NO_2$ anomaly divided by the mean background levels of $NO_2$ in each latitude/altitude bin) was less than 20% for most latitude/altitude bins. Therefore, the OSIRIS-measured decreases in $NO_2$ of up to ~ -30 to -50% after these volcanic eruptions are significant compared to background variability.

At both 0° (Figure 6) and 50°N (Figure 7), positive $NO_2$ anomalies are also observed, in particular before 2005. This is in part because background $NO_2$ is calculated using measurements taken for aerosol ratios less than 1.2. Below this threshold, levels of aerosol still vary, and tend to increase after ~2005 at most latitudes (see, for example, Figure 1). This leads to positive anomalies in $NO_2$ during periods with lower levels of volcanic aerosol, particularly before 2005. These positive anomalies are also apparent in the photochemical model results, which vary based entirely on aerosol levels. The photochemical model calculated background $NO_2$ is interpolated to the mean background aerosol as measured by OSIRIS over the full measurement period (e.g., the average aerosol extinction for January, when aerosol extinction ratios are below the threshold of 1.2). Therefore, early in the time series, the modelled $NO_2$ is calculated for lower levels of aerosol while background $NO_2$ is calculated for slightly higher levels of aerosol, leading to positive modelled $NO_2$ anomalies. There are some differences between the OSIRIS measurements and model results during periods with positive anomalies. For example, at 0° (Figure 6), positive anomalies are stronger in the OSIRIS data than in the model output, suggesting that the observed anomaly may not be fully explained by aerosol levels and may be related to other sources of variability.

At each altitude and latitude, the correlation coefficient between the OSIRIS percent difference in $NO_2$ and the aerosol extinction was calculated and is shown in Figure 8. Negative correlations between $NO_2$ and aerosol are observed at most

latitudes and altitudes in the lower stratosphere. Altitudes and latitudes that were affected by volcanic aerosol for at least one month in the time series are given by the magenta and green contour lines. For these altitudes and latitudes, the relationship between $NO_2$ and extinction tends to be stronger. In each latitude bin, the strongest correlation coefficient across altitudes is similar to the correlation coefficients for the partial VCDs and partial column AODs shown in Figure 5. In some cases, the strongest correlation coefficient within the profile is slightly lower than for the partial VCDs and partial column AODs, suggesting noise in either the $NO_2$ or aerosol extinction profiles or both.

## 5 Conclusion

Between 2002 and 2014, seven periods with enhanced volcanic aerosol were observed by OSIRIS at latitudes between 50°S and 80°N and, in most cases, were associated with reduced levels of $NO_2$ observed by both OSIRIS and MIPAS. For the partial column AODs between $5\text{-}7\times10^{-3}$, OSIRIS and MIPAS $NO_2$ VCDs decreased relative to background levels by ~20-50% and ~5-25%, respectively. For AODs greater than $7\times10^{-3}$, decreases in OSIRIS $NO_2$ reached ~40-60%. No MIPAS $NO_2$ measurements were available for AODs greater than $7\times10^{-3}$. MIPAS observed a smaller decrease in $NO_2$ than OSIRIS, which was found to be consistent with the effect of the MIPAS DOFS less than 1 at these latitudes.

The relationships between the percent differences in $NO_2$ relative to background levels and AODs are found for both OSIRIS and MIPAS, with correlation coefficient R between approximately -0.4 and -0.8 depending on the altitude and latitude range. Heterogeneous chemistry becomes saturated toward larger aerosol concentrations (e.g., Fahey et al., 1993) and can vary throughout the time series with other factors, such as temperature and available sunlight (e.g., Coffey, 1996), all of which can affect the linearity of the correlation . The variation of OSIRIS percent differences in $NO_2$ with partial column AOD was compared against photochemical model runs and was found to be consistent within estimated uncertainty.

A strong anticorrelation was observed between MIPAS $N_2O_5$ and OSIRIS AOD, with R ~ -0.9, however, no relationship was observed between MIPAS $HNO_3$ and OSIRIS AOD. The photochemical model suggests that increases in $HNO_3$ would be less than 10% for the observed partial column AODs, and therefore would be difficult to detect above other sources of variability.

The reductions in $NO_2$ observed in the present study would amount to less than 20% of the total column in the tropics and less than 10% of the total column toward higher latitudes, even for the largest aerosol events. This is much smaller than the column reductions of up to 50-70% observed after the Pinatubo and El Chichón volcanoes (e.g., Coffey, 1996), where the largest reductions in the total column of $NO_2$ occurred for periods with aerosol enhancements above 25 km (Koike et al., 1993), where the bulk of the $NO_2$ column resides. The results presented here are consistent with the smaller stratospheric aerosol loads and lower altitude range of the more recent volcanoes (e.g., Rieger et al., 2015).

## Appendix A: Relationship between aerosol extinction and aerosol surface area

Let $n(r)$ represent the number of particles, per unit volume, with a size between radius $r$ and $r+dr$, such that

$$N = \int_0^\infty n(r)dr \qquad\qquad (A1)$$

is the total number of particles per unit volume. The surface area density, $SA$, is then

$$SA = \int_0^\infty 4\pi r^2 n(r)dr \qquad\qquad (A2)$$

and the extinction, $k$, is

$$k = \int_0^\infty \sigma(r)n(r)dr \qquad\qquad (A3)$$

where $\sigma(r)$ is the extinction cross-section. It is convenient to use an extinction efficiency, $Q(r)$, such that $Q(r) = \sigma(r)/\pi r^2$ which represents the ratio of the extinction cross-section to the geometric cross-section. In this case

$$k = \int_0^\infty \pi r^2\, Q(r)n(r)dr \qquad\qquad (A4)$$

If it is assumed that $Q(r)$ can be replaced by an effective value, $\bar{Q}$, then

$$k = \bar{Q} \int_0^\infty \pi r^2 n(r)dr = \left(\frac{\bar{Q}}{4}\right) SA \qquad\qquad (A5)$$

where

$$\bar{Q} = \frac{\int_0^\infty \pi r^2 Q(r)n(r)dr}{\int_0^\infty \pi r^2 n(r)dr} \qquad\qquad (A6)$$

$Q(r)$ can be calculated for spherical particles using Mie theory; see, e.g., Hansen and Travis (1974). For a sulfate aerosol and a lognormal size distribution with $r_g$=0.08 µm and $\sigma_g$=1.6, $\bar{Q}$ =0.4. Since both surface area density and extinction are proportional to $r^2$, and the extinction efficiency is a weaker function of size than the absolute cross-section, it is advantageous to use this approach when the size information is less accurately known.

## Acknowledgments and Data

Thank you to Chris Roth for providing the principal components of QBO winds. This work was supported by the Natural Sciences and Engineering Research Council (Canada) and the Canadian Space Agency. Odin is a Swedish-led satellite project funded jointly by Sweden (SNSB), Canada (CSA), France (CNES), and Finland (Tekes). OSIRIS data are available at
http://odin-osiris.usask.ca. IMK/IAA-generated MIPAS data used in this study are available for registered users at http://www.imk-asf.kit.edu/english/308.php. BF was supported by the Spanish MINECO under grant ESP2014-54362-P.

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

Table 1: Parameters considered for model perturbation tests.

| Parameter | Model Settings | Perturbation(s) | Perturbation Comment |
|---|---|---|---|
| *SA* to *k* linear scaling factor | 10000 | Model Setting $\times$ 3; <br><br> Model Setting $\times$ 1/3 | Accounts for range in aerosol size distributions |
| Degree of *SA* to *k* non-linearity | $SA \propto k^{1.0}$ | $SA \propto k^{1.3}$; $SA \propto k^{0.7}$ | Following Thomason et al. (1997) |
| Ozone profiles | OSIRIS climatology (Bourassa et al., 2014) | Model Setting +10% | Estimate of uncertainty and variability in monthly mean |
| $NO_y$ profiles | From 3D model simulations (Olsen et al., 2001) | Model Setting +20% | Estimate of uncertainty and variability in monthly mean |
| Temperature | Climatology from Nagatani and Rosenfield (1993) | Model Setting +5 K | Estimate of uncertainty and variability in monthly mean |
| Surface albedo | 0.1 | 0.3 | Difference between mean, effective albedo with and without clouds |
| NO + $O_3$ reaction rate coefficient | JPL Publication 15-10 (Burkholder et al., 2015) [A=$3\times10^{-12}$; E/R=1500] | Model Setting +10% | Estimated uncertainty from Burkholder et al. (2015) |
| $NO_2$ absorption coefficient | JPL Publication 15-10 (Burkholder et al., 2015) <br><br> [Table 4C-2] | Model Setting +10% | Estimated uncertainty from Burkholder et al. (2015) |

Table 2: Conversion factors for aerosol extinctions measured at various wavelengths.

| Conversion (nm) | Angstrom coefficient | Conversion factor (ratio of extinctions) |
|---|---|---|
| 1020 → 525 | 2.5 | 5.42 |
| 750 → 525 | 2.3 | 2.27 |
| 1020 → 750 | 2.8 | 2.39 |

Table 3: Summary of volcanoes observed in OSIRIS partial column AOD and associated OSIRIS NO₂ VCDs. References given in footnotes describe stratospheric aerosol following these eruptions. Note that this table lists eruptions that were followed by significant increases in OSIRIS aerosol extinction. Therefore, it does not include all volcanoes known to have affected the stratosphere during this time-period.

| Volcano Name[a] | Eruption Date | Eruption Latitude | Extent of aerosol enhancement observed by OSIRIS[b] | Effect on OSIRIS-observed NO₂ partial VCD |
|---|---|---|---|---|
| Manam[c] | 27 Jan 2005 | 4°S | Both hemispheres, confined to the tropics | Minimal effect on observed NO₂ |
| Soufrière Hills[d] Rabaul (Tavurvur) | 30 May 2006 7 October 2006 | 17°N 4°S | Both hemispheres, reaching high latitudes in spring 2007 | For 50°S-0° and 40°-60°N, NO₂ is lower by ~10-40% |
| Mt Okmok[e] Kasatochi[e] | 12 Jul 2008 7 Aug 2008 | 53°N 52°N | Combined effect of both volcanoes, reached tropics in Dec 2008-Jan 2009 | For 40-80°N NO₂ is lower by ~20-40% |
| Sarychev Peak[f] | 12 Jun 2009 | 48°N | Large AODs until Dec 2010, mostly confined to northern hemisphere mid-latitudes and high latitudes | For 30-80°N, NO₂ is lower, reaching reductions of up to ~45-55% for 40-80°N |
| Mt Merapi | 4 Nov 2010 | 7°S | Both hemispheres, small signal confined to tropics in northern hemisphere and extending to higher latitudes in southern hemisphere in Jan 2011 | Minimal effect on observed NO₂ |
| Nabro[g] | 12 Jun 2011 | 13°N | Large AODs throughout the northern hemisphere until Jan 2012, with smaller AODs until Jun 2012 | 20-80°N, NO₂ is lower, reaching reductions of up to ~50-55% for 50-80°N |
| Kelut | 13 Feb 2014 | 8°S | Both hemispheres, confined to the tropics | For 10S°-0° NO₂ is lower by ~20-40% |

[a] Eruption dates and latitudes from (Höpfner et al., 2015) and reports available at http://volcano.si.edu/.

[b] OSIRIS does not measure AOD in the winter hemisphere and therefore may not capture the full extent of aerosol enhancement.

[c] (Bourassa et al., 2012b)

[d] (Prata et al., 2007)

[e] (Bourassa et al., 2010; Kravitz et al., 2010; Sioris et al., 2010)

[f] (Haywood et al., 2010; Jégou et al., 2013; O'Neill et al., 2012)

[g] (Bourassa et al., 2012a)

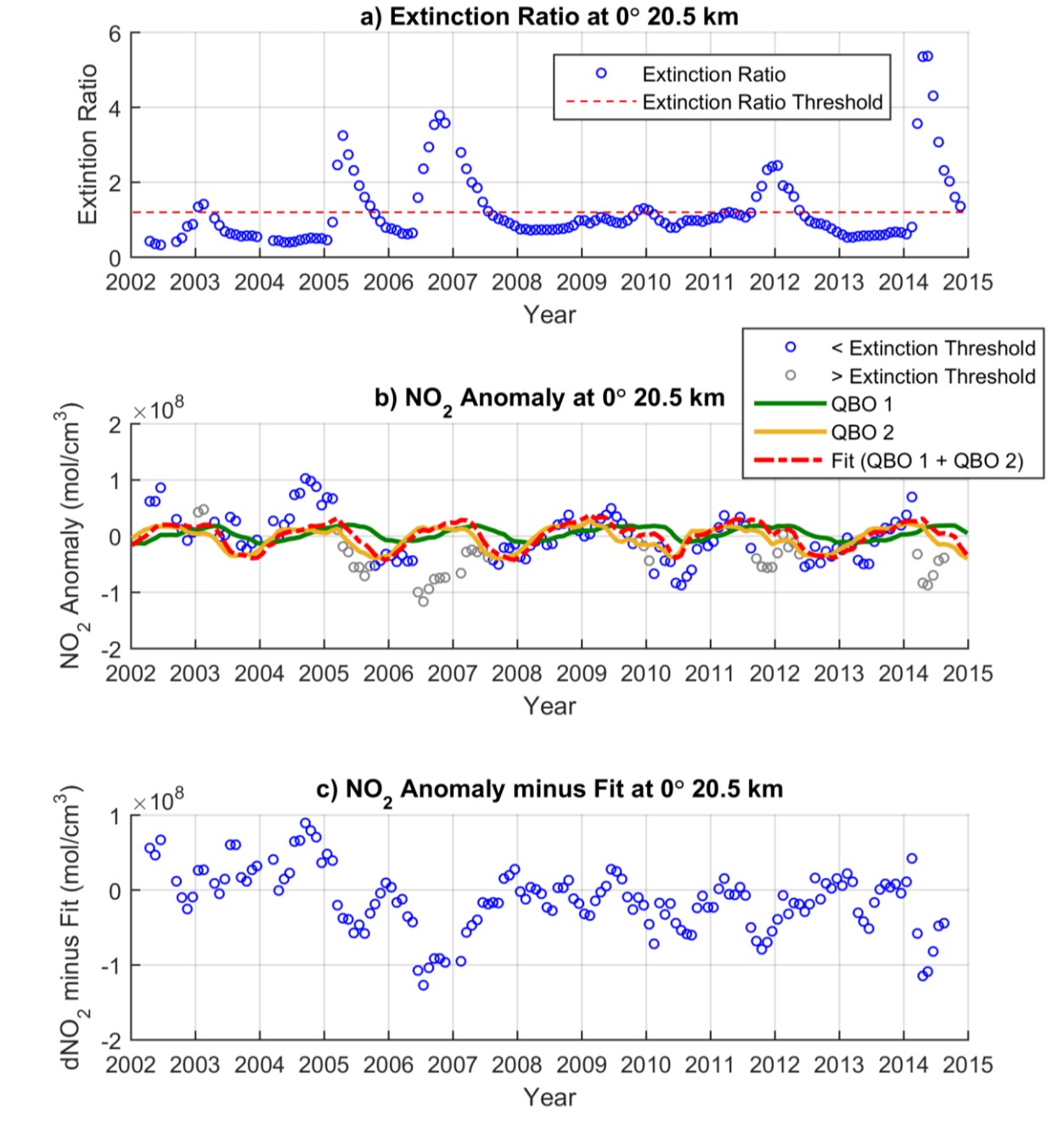

Figure 1: QBO fitting results for 0° latitude at 20.5 km. (a) Extinction ratio time series (blue circles) with extinction ratio threshold (red dashed line). (b) NO₂ anomaly time series for time periods with extinction ratio less than extinction ratio threshold (blue circles) and time periods with extinction ratio greater than extinction ratio threshold (grey circles). The fits for

the two QBO principal components – QBO 1 (green line) and QBO 2 (yellow line) – as well as the total fit (red dashed lined) to the NO$_2$ anomaly time series during time periods with extinction ratio less than extinction ratio threshold are also shown. (c) NO$_2$ anomaly time series after subtraction of the fit.

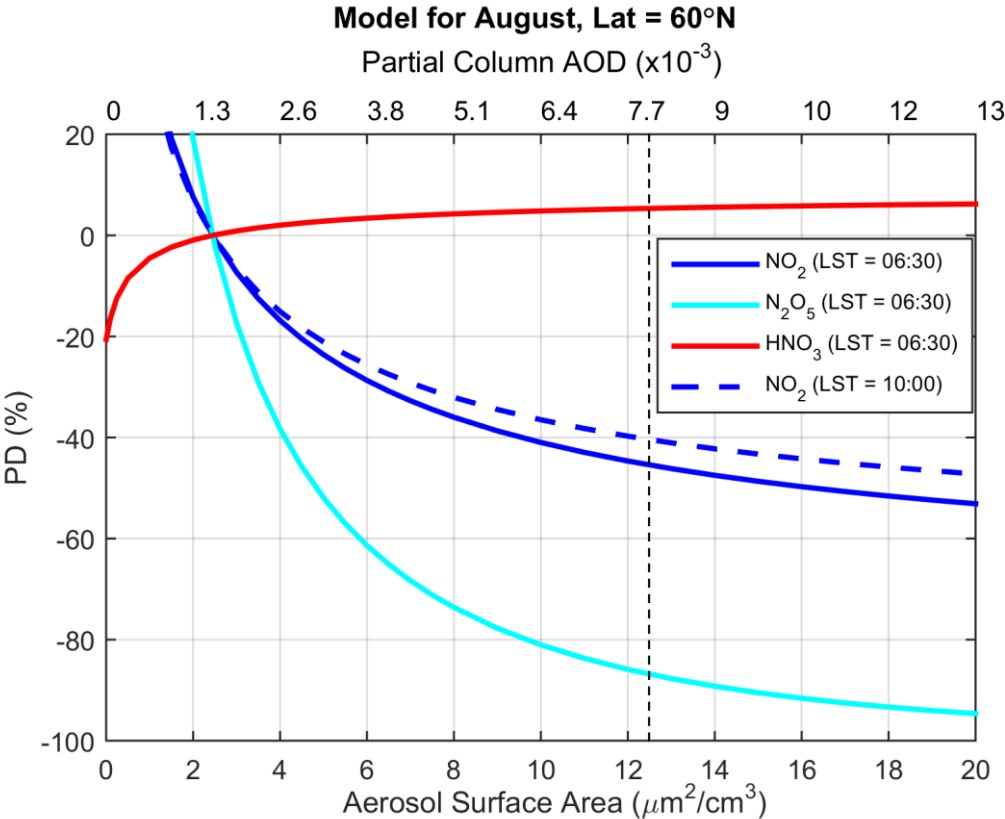

Figure 2: Modelled variations of NO$_2$, N$_2$O$_5$, and HNO$_3$ with stratospheric aerosol at 60°N in August. Aerosol surface area (bottom x-axis) and partial column AOD (top x-axis) are both shown. The y-axis gives the percent difference (anomaly – background)/background for partial VCDs of NO$_2$ (blue line), N$_2$O$_5$ (cyan line), and HNO$_3$ (red line) at the approximate OSIRIS local time (06:30 LT), and for NO$_2$ (blue dashed line) at the approximate MIPAS measurement time (10:00 LT). AODs and VCDs were calculated for a 5 km layer, starting ~3 km above the tropopause. The black dashed line indicates partial column AOD = 8x10$^{-3}$, approximately the largest value in the OSIRIS measurements.

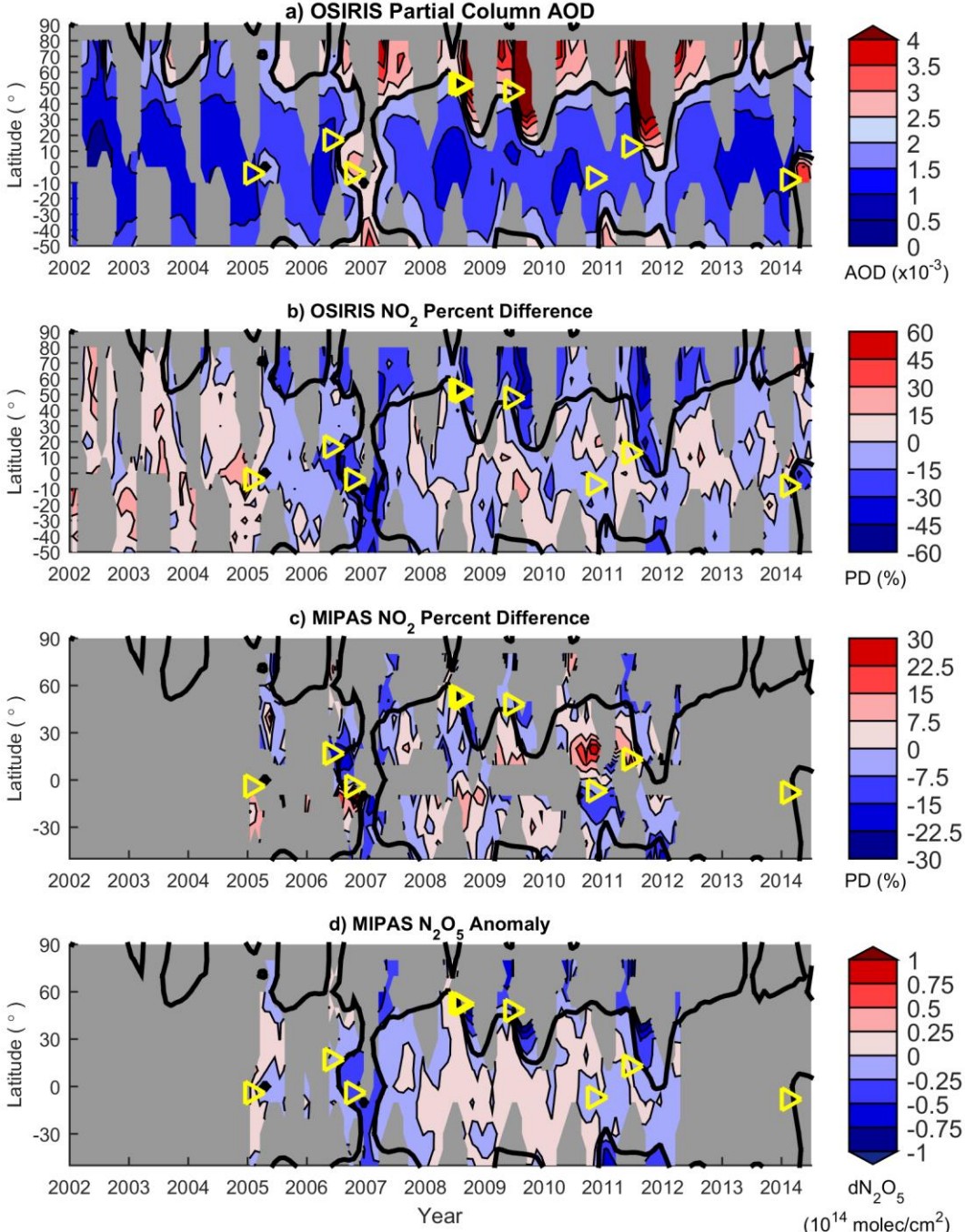

Figure 3. Contour plots of time (x-axis) versus latitude (y-axis) versus (a) OSIRIS partial column AODs, (b) percent difference of OSIRIS $NO_2$ VCDs, (c) percent difference of MIPAS $NO_2$ VCDs, and (d) MIPAS $N_2O_5$ anomaly. The partial column AODs and VCDs are calculated for 5 km layer starting ~3 km above the thermal tropopause. Percent differences are relative to background levels of $NO_2$. Note that different colour-scale ranges are used for OSIRIS and MIPAS $NO_2$ percent differences. The thick black contour lines show OSIRIS AOD $= 2 \times 10^{-3}$, extrapolated to latitudes/times where data were unavailable. The yellow triangles indicate the volcanic eruptions that were followed by significant increases in OSIRIS aerosol extinction, as listed in Table 3.

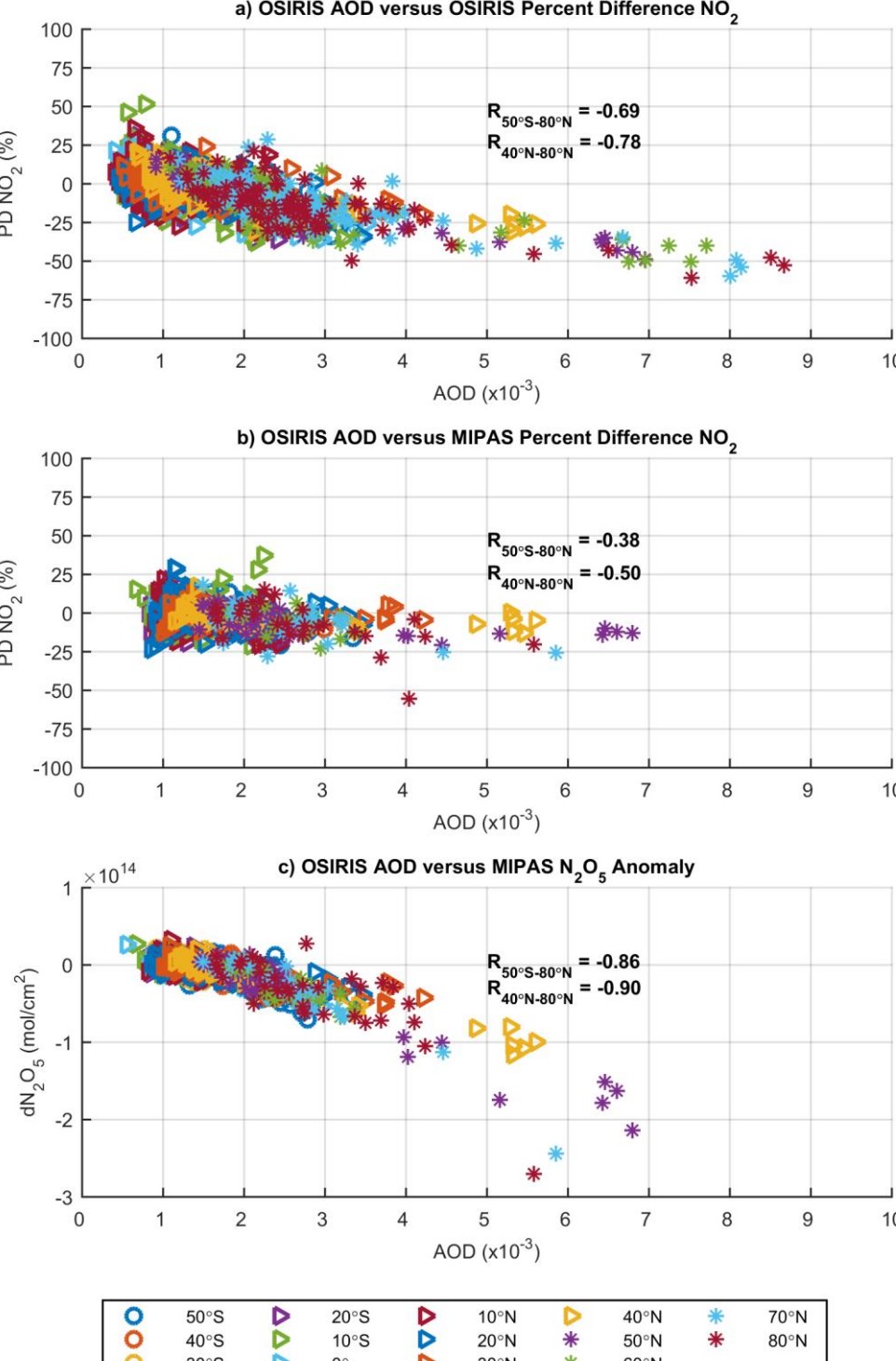

a) OSIRIS AOD versus OSIRIS Percent Difference NO$_2$

$R_{50°S-80°N}$ = -0.69
$R_{40°N-80°N}$ = -0.78

b) OSIRIS AOD versus MIPAS Percent Difference NO$_2$

$R_{50°S-80°N}$ = -0.38
$R_{40°N-80°N}$ = -0.50

c) OSIRIS AOD versus MIPAS N$_2$O$_5$ Anomaly

$R_{50°S-80°N}$ = -0.86
$R_{40°N-80°N}$ = -0.90

| | | | | |
|---|---|---|---|---|
| 50°S | 20°S | 10°N | 40°N | 70°N |
| 40°S | 10°S | 20°N | 50°N | 80°N |
| 30°S | 0° | 30°N | 60°N | |

Figure 4. Scatter plots of OSIRIS AOD versus (a) OSIRIS and (b) MIPAS percent difference of $NO_2$ VCD relative to background levels and (c) MIPAS $N_2O_5$ anomaly. The legend shows the measurement latitude. R for $NO_2$ percent difference and AOD for data collected between 50°S and 80°N, and 40°N and 80°N are given in the plot. The p-values for all calculated R are less than $1x10^{-8}$.

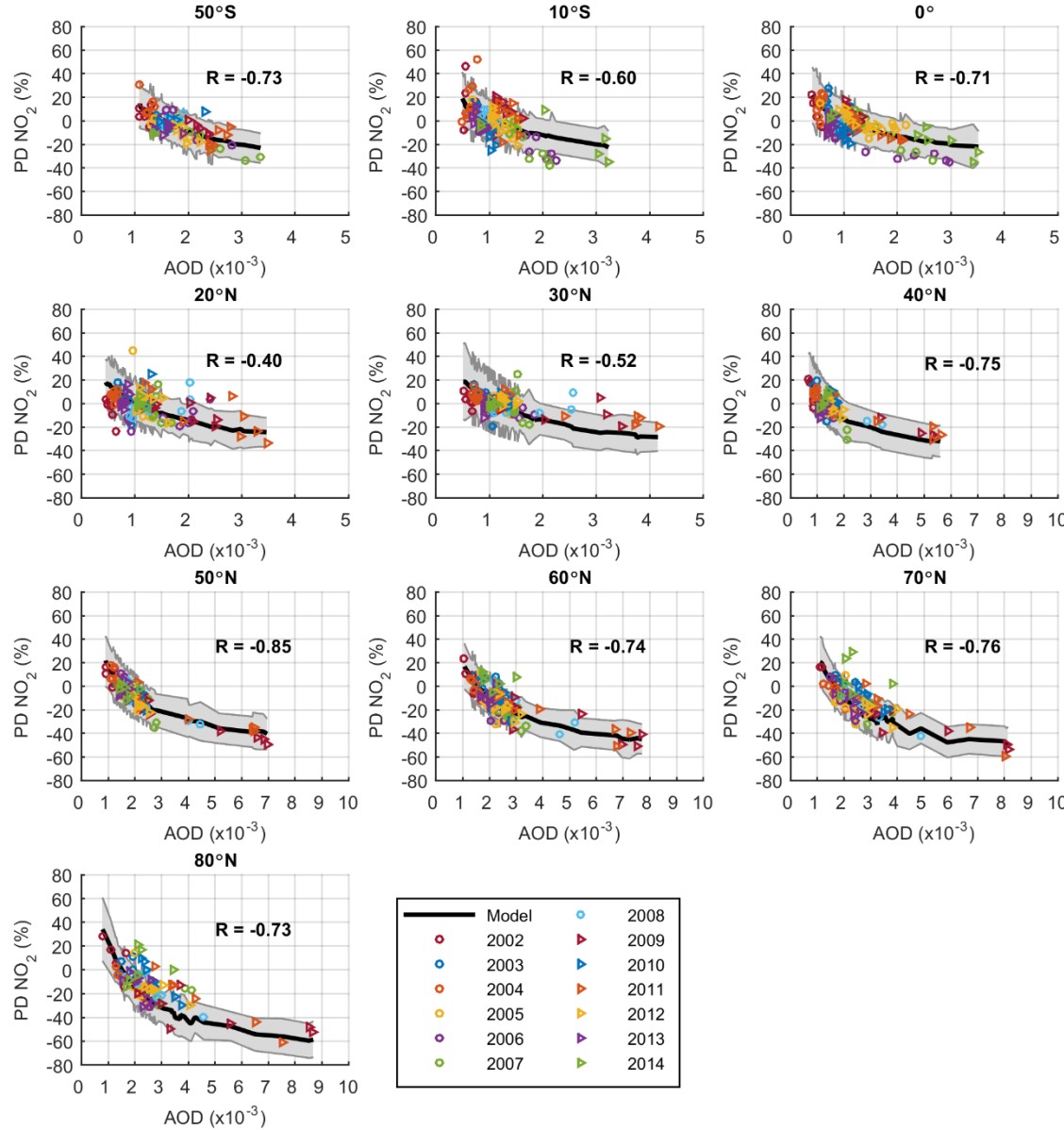

Figure 5. Scatter plots of OSIRIS AOD versus OSIRIS $NO_2$ percent difference in VCD relative to background levels for various latitudes, with R given in the plot. The legend shows the measurement years. Modelled values are shown with black line, with shaded region representing uncertainties in aerosol extinction to aerosol surface area conversions and model input

parameters, as described in Sect. 2.3. The p-values for R in all panels are less than $1\times10^{-5}$.  Note that the scale of the x-axis varies between panels so that the data can be seen more clearly.

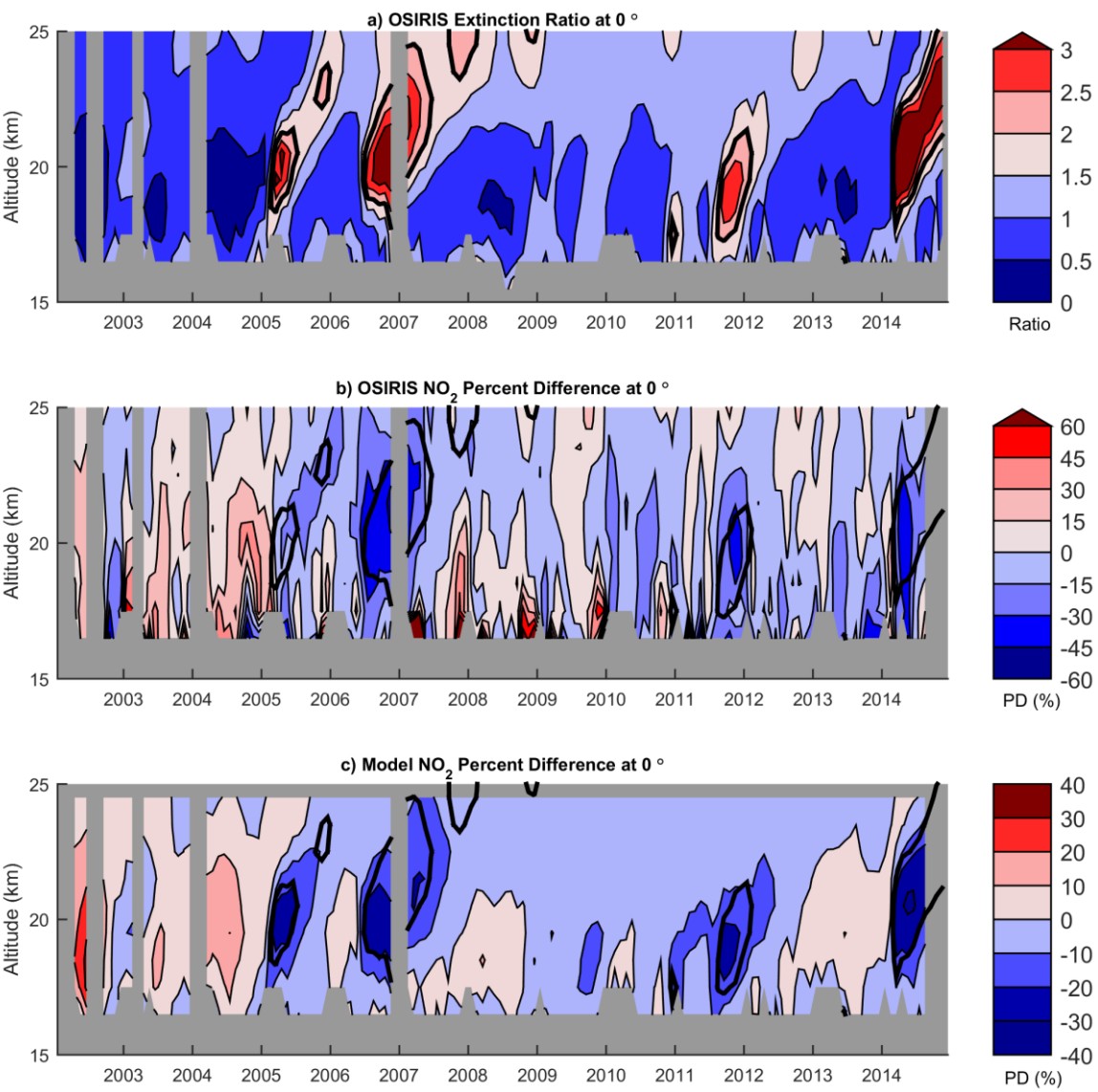

Figure 6. Profiles of aerosol extinction ratio and $NO_2$ percent difference at 0°.  Time series of (a) OSIRIS extinction ratio, (b)
5    OSIRIS percent difference $NO_2$ relative to background levels, and (c) modelled percent difference $NO_2$ relative to background

levels. Note that different colour-scale ranges are used for the OSIRIS and model percent differences. The thick black contours are for extinction ratios = 2.

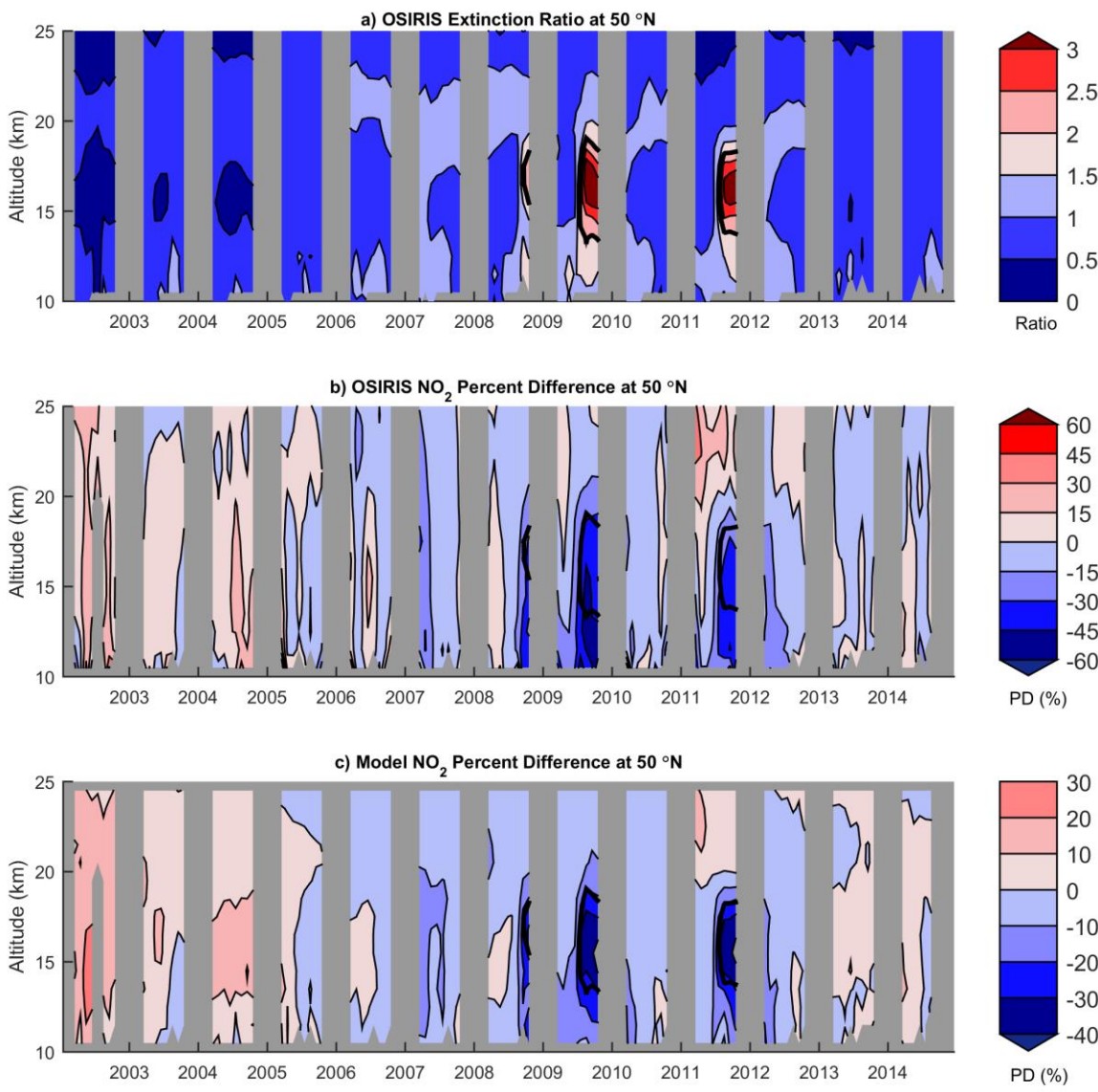

Figure 7. Profiles of aerosol extinction ratio and $NO_2$ percent difference at 50°N. Time series of (a) OSIRIS extinction ratio, (b) OSIRIS percent difference $NO_2$ relative to background levels, and (c) modelled percent difference $NO_2$ relative to

background levels. Note that different colour-scale ranges are used for the OSIRIS and model percent differences.   The thick black contours are for extinction ratios = 2.

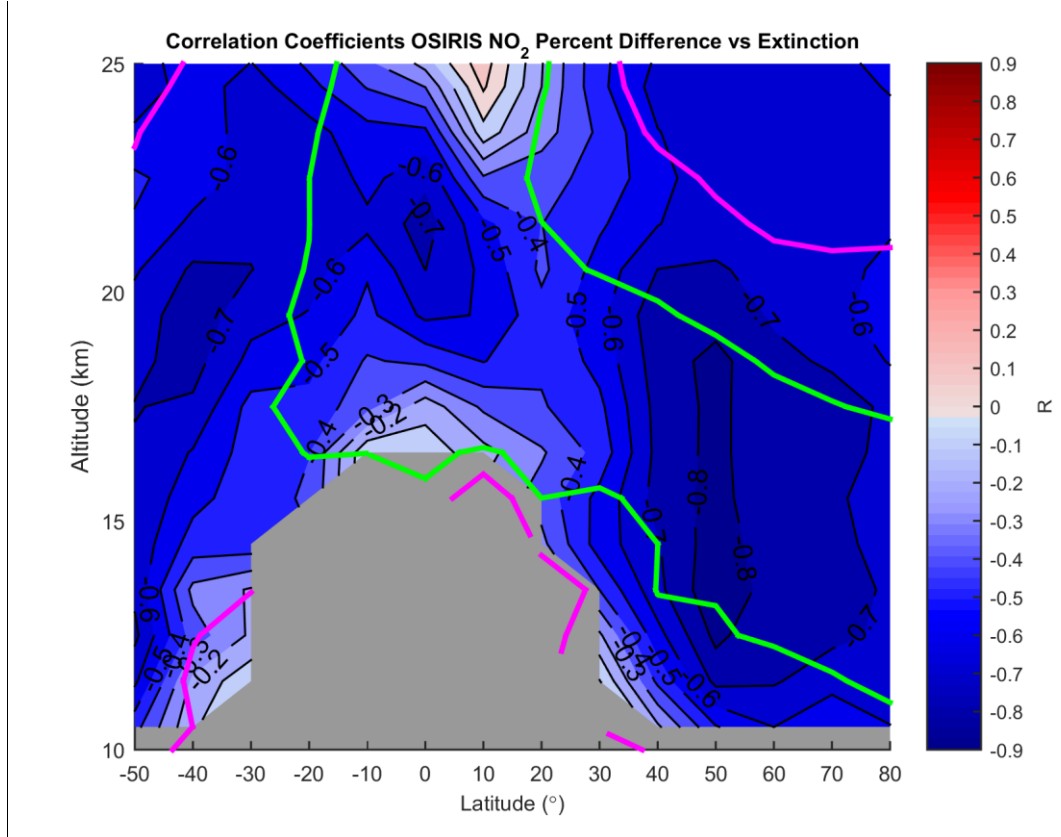

Figure 8. Correlation coefficient for time series of OSIRIS percent difference in $NO_2$ versus aerosol extinction for each latitude and altitude, the magenta and green contours are for latitudes and altitudes at which maximum extinction ratios over the time series was 1.2 and 2, respectively.