# Peer review of "Effect of volcanic aerosol on stratospheric NO2 and N2O5 from 2002-2014 as measured by Odin-OSIRIS and Envisat-MIPAS"

_Atmospheric Chemistry and Physics, 2016_

## Referee Comment (RC1) · Anonymous Referee #3 · 1 Jul 2016

**Referee report**

**1. Effect of volcanic aerosol on stratospheric NO$_3$ and N$_2$O$_5$ from 2002-2014 as measured by Odin-OSIRIS and Envisat-MIPAS**

C. Adams, A.E. Bourassa, C.A. Mc Linden, C.E. Sioris, T. von Clarmann, B. Funke, L.A. Rieger, and D.A. Degenstein

**General comments:**

This work investigates the relationship between aerosol load enhancements after a volcanic eruption and the anomalies observed in the NO$_2$ vertical partial column over the 3-7 altitude range, as observed by OSIRIS and by MIPAS. The study reveals quite robust correlations between the aerosol optical depth and the NO$_2$ anomalies, especially when comparing between OSIRIS aerosol and NO$_2$ features, but the results are much less convincing while comparing with MIPAS NO$_2$ anomalies. NO$_2$ anomalies in presence of polar stratospheric clouds, a case for which discussion is avoided in the paper, seems not to reflect the denitrification expected in this case.

Overall, the paper provides interesting observations, but I think that two aspects have not been considered appropriately and might explain the inconsistencies found between MIPAS and OSIRIS measurements.
First, the authors consider monthly zonal means, with 10° latitude bins. This is a very coarse grid to study volcanic plume such as the ones considered in this study. The effects the authors want to study are largely diluted in the bin averaging and the mean values obtained from the binning are likely to be biased depending on the coverage of the instrument. Obviously, the effect of the bias is expected to increase if the comparison concerns data from two different instrument (with different coverage).
Another similar source of bias could be the use of climatological data for ozone, neglecting the effects of local chemistry
The second weakness of the paper, in my opinion, concerns the discussion of the comparison between OSIRIS and MIPAS data, where arguments based on the limited degrees of freedom provided by MIPAS are used to justify the disagreement between results obtained from both instruments. Before drawing definitive conclusions from this hypothesis, the authors should check their interpretation by degrading both datasets to similar resolutions.

These issues should be addressed before publication of the manuscript.

**Specific comments:**

**Abstract**

- L. 23: The authors mention percent difference of up to ~25%. They should mention with respect to what (to OSIRIS? To quiescent periods?)

**1. Introduction**

- L. 8, p.1: The authors should define $NO_x$ like they defined $NO_y$. Possibly at the same place to lighten the text, see technical comment on L. 4.

- L. 16, p.2: For the sake of clarity: "Nox/NOy decreased with increasing aerosol surface area".

**2. Satellite and model datasets**

**2.3 Photochemical modelling**

- L. 16, p.4: Which type of climatological data are used for ozone and temperature? And why don't they use MIPAS ozone data which would be fully consistent with the used data for $SO_2$, $N_2O_5$, $HNO_3$ and $NO_2$ ? Perliski et al. (1989), cited in Randeniya et al. (op. cit., 1997), indicate that the ozone behaviour at high latitudes is expected to be dominated by local chemistry during summer, which is the time and region of interest for the present study, and where Eq. (2) is expected to have the greatest impact on ozone. Hence, using climatological data for the ozone field in this specific study focussing on the effect of volcanic eruptions might bias the results of the analysis.

- L. 18, p.4: I don't understand what the authors mean with "but fixed to a specified Julian day". Please clarify.

- L. 20, p.4: Since Thomason's climatology covers the whole period 1979-1995, it would be useful to explain in a few words which data are used in the present study (specific year and/or region?). If they only use the approximated expressions (5) of Thomason's work as discussed in the following of the section, the authors could already announce that here (e.g.: "(…) using the aerosol surface area climatology of Thomason et al (1997) as explained later, (…)"). Actually, I don't know why the authors want to consider heterogeneous chemistry on background stratospheric aerosols, since they aim at studying typically volcanic situations. Besides, they mention further that they match the extinction coefficient with OSIRIS values, which should normally reflect this volcanic feature. All these points should be clarified.

- L. 24, p.4 to L. 10, p.5: The estimation of the SA in function of the extinction seems particularly crude, with a succession of approximations with choices which are not always clear nor convincing.

  The authors use the fact that in the case of Kasatochi, the mean particle size (see also next comment) decreases and that in the case of Sarychev, it increases, to choose a dependence between SA and the extinction which reflect none of these cases. This is a questionable way to approach this investigation focussing on a selection of recent volcanic eruptions (including those two ones). It is also worth to mention that Thomason et al (1997, op. cit.) uses such linear expression to

describe cases where the extinction is higher than 2. $10^{-2}$ km$^{-1}$, which is a really high value rarely encountered in the period considered here. The wavelength used to characterize the extinction in the equation in L. 26, p.4 is not mentioned, making it meaningless and preventing to compare this value with Thomason's coefficient (equal to 2000 to characterize the extinction at 1020 nm). They also "test" a non-linear SA dependence mentioning Thomason's work, but their choices are quite different form what Thomason proposes. The choice of p=0.7 is quite similar to Thomason's one in the case of the weakest extinction values, but the choice p=1.3 is much higher than the value of 1 used with the highest extinction values.

The final scaling "to account for potential errors" gets rid for good of any reference to real cases, making the reader definitively lost in the accumulation of assumptions.

- L. 5, p.5: Sioris et al. (2010) don't claim that the stratospheric particle size decreases in the case of Kasatochi. They consider statistical values (through median radius and Angström exponent) which show a decrease of the averaged particle size. This is not the same thing. A particle size decrease supposes the occurrence of some evaporation process, while what happens here is the addition of a significant amount of very thin particles. The authors should change their formulation to avoid the confusion.

- L. 1, p. 5: The reference by Hansen and Travis, 1974 is a very interesting general reference on light scattering, but I don't see anything in this reference able to clarify the choices and formulation of the equation in L. 26, p.4. Hence, I am not sure it is really useful here.

**3. Calculation of monthly averages, anomalies, and baseline levels**

**3.1 OSIRIS and MIPAS**

- L. 13-20, p. 5: The use of monthly zonal means over latitude intervals as large as 10° might really bias the data and limit the quality of the correlative study of quantities derived from two different experiments such as OSIRIS and MIPAS in the present case. If as few as 5 measurements are possible for one bin, it could be possible, for instance, that one instrument covers only regions with low aerosol background while the other one catches the plume of an eruption. Or that one instrument covers a large region of the bin and returns average values of a quantity while the other one only catches some very local spot with specific (low or high) volcanic load. Did the authors prevent this kind of situation in some way?

- L. 1 p. 7: Concerning the NO2 response to the QBO, do the authors mean the fitted response as illustrated by the cyan curve in Figure 1b?

- L. 11-15, p. 7: same remark as for L. 13-20, p. 5: I suppose that the model covers the whole 10° latitude monthly bin, while corresponding measurements might cover a reduced region of the bin, introducing potentially a bias in the analysis.

**3.3 Conversions between partial column AODs, aerosol extinction, and extinction at various wavelengths**

- L. 21-24, p. 7 (or L. 15-18, p. 5): I guess that profiles for which valid data don't cover the whole interval 3-7 km above the tropopause are rejected. This could be mentioned for the sake of completeness.

**4. Results**

**4.1 $NO_2$, $N_2O_5$, and $HNO_3$ VCDs**

- L. 26, p.8 and Figure 3: Overall, there is indeed a very clear correlation between the AOD enhancement and the $NO_2$ anomalies found by OSIRIS in the Northern latitudes. On the contrary, in the Southern polar latitudes, most of the time, no significant $NO_2$ decrease is found by OSIRIS and even more, strong local enhancements are observed each year. In the case of the Sounthern polar latitudes, the high AOD is most probably due to PSC, for which one would expect a denitrification, thus a decrease in $NO_2$. MIPAS $NO_2$ seems rather to behave in the opposite way, although the very limited coverage of the "volcanic regions" defined by the cyan curves would impose to be cautious in any conclusion.

- L. 2, p.9: I don't understand the reason given by the authors why the AOD should be excluded from the correlation before of its large variability. Large volcanic eruptions are able to produce an even larger variability.

- L. 6-7, p.9: Again, the authors consider monthly zonal bins with 10° latitude intervals for their analysis. This might be the reason for the apparently inconsistant behaviour (MIPAS vs OSIRIS, Northern latitudes vs. Southern latitudes) shown in Figure 3 (see comment on L.26, p.8 and Figure 3). Although the use of monthly zonal means is widely used in the community, as mentioned above, such large intervals are not well suited to study atmospheric processes at the level of a volcanic plume for such kind of eruptions, because the spatial extend and the temporal duration of the volcanic perturbation is relatively limited with respect to the spread of the bins. Hence, biases are potentially important and make any conclusion uncertain, especially if different instruments with different coverage and data rates are compared. The authors should remake their analysis by considering much shorter time and latitude intervals to see if the inconsistencies persist.

- L. 9-15, p.9: The validity of the explanation given by the authors could be easily checked by degrading the OSIRIS using the MIPAS' averaging kernels (and possibly vice-versa). This way would allow comparing like with like. Did the authors make such check?

- L. 29, p.9-L. 1, p. 10: The concept "somewhat linear" is strange. Overall, the interpretation of the shape of the curve is highly subjective, and the superposition of the modelled values on the plots biases our perception. As an example, the OSIRIS plots at 20°N show a behaviour which is neither linear at low AOD, nor

saturated at the highest AOD values. The authors should remove these dubious interpretations.

- L. 4-5, p.10: In the same way, the agreement in shape and in quantity between modelled and observed data is very relative, and in some case, quite bad. Hence, the authors should qualify their affirmation.

**4.2 OSIRIS NO$_2$ profiles**

- L. 25, p.10: The authors should remove the sentence "At this latitude, decreases in NO$_2$ are observed between ~10-20 km". This sentence is confusing, since in some cases (e.g. in 2002), an increase is observed in NO$_2$ instead of a decrease, and anyway, the next sentence expresses appropriately and in more detail what the authors mean.

**5. Conclusions**

- L. 20-21, P. 11: I think the conclusion concerning the influence of the DOFS on the disagreement between MIPAS and OSIRIS is premature, and should be verified as proposed above, before it is claimed.

- L. 22, p. 11: This line, with the qualification of "somewhat linear", should definitely be removed. The characterization of this relationship using the correlation coefficient is more than sufficient. In the same way, the expression "perfect linearity" on the next line should also be removed. As long as observations are concerned, there cannot be any perfect linearity.

- L. 27-28, p.11: The last sentence should be qualified according to the comment in L. 4-5, p.10.

**Technical corrections:**

- L.19, p.1: The authors could consider using "relationship" in the singular, or more precise, the word "correlation"?

- L.23, p1: "periods affected by volcanic aerosol"

- L. 4, p.2: reformat the parenthese after BrONO$_2$ (no subscript). Writing "NO$_y$ species (where NO$_y$ = NO$_x$ +HNO$_3$+etc.+BrONO$_2$, and NO$_x$=etc.; e.g. Coffey 1996)" might be more fluent. See also specific comment on L. 9.

- L. 13, p.4: I suggest to write 10:00 LT to be consistent with the previous mention of time, and for the sake of clarity.

- L. 6, p.5: "to keep the scattering efficiency constant".

- L. 15, p.8: It seems there is a problem of cross reference for Table 2.

- L. 13, p.10: "time [blanco] series"

---

## Referee Comment (RC2) · Anonymous Referee #2 · 8 Jul 2016

**1   General Comments**

The paper addresses the reduction of NO$_2$ and N$_2$O$_5$ by heterogeneous reactions in the lower stratosphere after mediumsize volcanic eruptions based on satellite data. Anticorrelations between aerosol optical depth and NO$_2$ can be seen. A big problem with the paper is that it totally relies on monthly zonal means which is not appropriate for the rather local and short lived volcanic plumes. Because of this, MIPAS SO$_2$ data are provided as 5-day means (Höpfner et al., 2015). Several important volcanic events listed there are missing or placed at the wrong latitude (Figure 3). There is also no need to use inconsistent climatological ozone for the photochemistry, from both instruments,

OSIRIS and MIPAS, are selfconsistent data available. The paper presents interesting data but before publication it needs major revision.

**2  Specific comments**

Abstract: Do the authors mean a 4 km thick layer above the tropopause, in tropics and midlatitudes, and against what conditions is the change?

Please define $NO_x$ in line 9 of page 2.

Please improve wording in line 13 of page 3, it contains contradictions.

From the data version number, it looks like that Höpfner et al (2015) is used (line 9ff, page 4), here also the $SO_2$ data prior to 2005 are OK. In these data, especially if the 5-day means are used, all important volcanic events should be identified with significance (see also lines 15 and 25, page 8). This is not the case for the older dataset presented in Höpfner et al (2013) which had the focus on the middle and upper stratosphere.

For the crude assumptions on the Mie scattering efficiency the wavelength should be repeated (line 2, page 5). The statements on particle size (line 5, page 5) are confusing, more details please, give at least a range for the effective particle size. If you model particle size from aerosol formation from injected $SO_2$, you get in increase in effective particle size for both volcanoes. What is the basis for the crude error assumptions (factor 3)?

Isn't there also an averaging kernel for OSIRIS (line 19ff, page 5)?

In Fig. 1 an additional panel with the zonal wind at 20 km (?) might be useful (line 21, page 6).

Why are different partial columns given in section 3.3 (line 23, page 7) and earlier in the text (including abstract)?
In Table 2 at least the eruption of Rabaul in Oct. 2006 is missing.

There appear to be contradictions between Fig. 3 and 4. Improve Fig. 3 concerning $SO_2$ with the Höpfner (2015) 5-day dataset. Place the symbols for the volcanoes at the correct latitude. It might be better to use volume mixing ratios at 19 km (or 3 km above the tropopause) instead of the partial columns to reduce data gaps. The results are also sensitive to the treatment of negatives in the individual data.

I don't understand the statement on tropospheric water vapor (line 22, page 8). The current understanding is that for explosive eruptions $SO_2$ is directly injected into the stratosphere, in the plume only water from the volcano might matter, but the satellite sees only what comes out of the plume.

Section 4.2: In Figs. 1, 6 and 7 appear often extinction ratios < 1. Please explain or correct, from definition this should not happen. Please adjust color bars to reasonable range,

Don't use formulations like 'somewhat linear' (line 4, page 10; line 3, page 12).

**3 Technical corrections**

Line 20ff, page 1: Better 'anticorrelation' instead of 'relationship'.

Line 6, page 5: typo and bad wording.

Figure 3, caption line 4: Do you mean a 4 km thick layer 3 km above the tropopause? Please improve text.

Truncate Figs. 6 and 7 at 12 km, the data below are not relyable. Say 'aerosol extinction ratio' also in captions. The black contours are superfluous. The colorbars should have the same steps as the colors in the figures (less is more!).

Fig.8: Caption: Say 'correlation coefficient' instead of 'R'. The colorbar should have the same steps as the colors in the figure.

[Figure]

---

## Referee Comment (RC3) · Anonymous Referee #2 · 8 Jul 2016

Caution, the line numbers in the report do not refer to the published version. They can be shifted by more than 10 (backward).

---

## Referee Comment (RC4) · Anonymous Referee #1 · 13 Jul 2016

This work investigates the effect of volcanic aerosol on NO2 and N2O5 in the stratosphere. The study is based on the analysis of OSIRIS and MIPAS measurements, and on photochemical box model runs. It shows that enhanced aerosol optical depth following volcanic eruptions between 2002 and 2014 is generally associated with negative NO2 and N2O5 anomalies.

The manuscript presents interesting and novel results, but important issues should be addressed before publication. The main one, in my opinion, is that the analysis relies on averages calculated into monthly 10° latitude bins. Using such large intervals in time and space to study quite short-lived and local events such as volcanic plumes can lead to important biases in the results. Please see some questions and suggestions

below.

**Specific comments:**

p.2, l.9: "NOx" should be defined.

p.4, l.18: Please give more details on the climatological profiles used for ozone and temperature, and explain your choice of using these profiles rather than using directly MIPAS and OSIRIS measurements of O3 and T.

p.4, 19-20: Please clarify the sentence "All remaining species are calculated to be in a 24-hour steady-state by integrating the model over 30 days, but fixed to a specified Julian day".

p.4, l.22: The effects of polar stratospheric clouds are not considered here. However, these could play a role in the altitude range under consideration, especially in summer, which is the season on which this study is focused. This should be discussed while interpreting the results.

p.5, l.9-10: How the factors 3 and 1/3, used to account for potential errors in your background aerosol surface area, have been chosen?

p.5, l.21: Place clarify "for the five measurement layers from 3-7 km above the NCEP thermal tropopause..." I do not clearly understand which measurement layers you are using.

Section 3: The analysis is based on monthly means calculated in 10° latitude bins. The choice of such large intervals in time and space does not seem to be the most appropriate to study volcanic plumes, which are quite short-lived and local events. A different distribution of the observations from OSIRIS and MIPAS in a given bin might lead to very different results. The same comment is also relevant for the photochemical model, which covers the whole bin in a uniform way, while this is probably not the case for the observations. Was it not possible to perform the analysis using more appropriate bins? In this case, the authors should find a way to estimate the sensitivity of their results to this problem, and this should be thoroughly discussed in the paper. Have the authors try to use only the modelled data in collocation in time and space

with the observations?

p.8, l.13-17 and Fig.3: Please comment the high AOD levels at southern high latitudes, which are obviously not due to volcanic eruptions. We can read, p.9 l. 5-6, that these are "perhaps due to polar stratospheric clouds". This should be discussed already in the description of Fig.3, and this statement should be explained, or at least associated with a reference.

p.9, l8: Is the second interval considered to calculate the correlation coefficients 40°N-80°N or 40°S-80°N? The information given in the figure is inconsistent with what is said in the text. Same remark p.9, l.22.

p.9, l.16-18: This could be checked by applying MIPAS averaging kernels to the OSIRIS data. This has already been done in several studies comparing MIPAS data to other data sets characterised by a better vertical resolution.

p.10, l.15-16: Was the vertical resolution of the MIPAS N2O5 profiles also not good enough to look at the effect of volcanic aerosol on this species as a function of altitude?

**Technical corrections:**
p.1, l.24: Remove "of"
p.1, l.25: Please change "relationship" to "anti-correlation"
p.3, l.22, p.4, l.1, p.4, l.14 for example: Please write "lower than" or "greater than" instead of "<" or ">".
p.5, l.19: For the sake of clarity, please add "number densities" after "SO2, NO2, N2O5 and HNO3".
p.6, l.13: "on the same order of magnitude AS the variation..."
p.6, l.27: "...periods that WERE not affected by volcanic aerosol."
p.10, l.17: Please add "percent difference" in "OSIRIS and modelled NO2 percent difference profiles".
p.10, l.20-23: For the sake of clarity, the given percent different values should be negative (same remark p.11, l.2)
Fig. 3, 6, 7 and 8: These figures would be clearer if there was a limited number

of colours in the colour bar, so that it corresponds to the colour levels shown in the plots.

---

## Author Comment (AC1) · 10 Apr 2017

**Response to Referee 3**

We thank you for your comments, which have helped to improve our manuscript. Below we address the recommended changes point-by-point.

**General comments:**

Since these general comments have been repeated/clarified in the specific comments below and by other reviewers, we have labelled them A, B, C (as listed below), so that the responses could be more easily cross-referenced.

*A. NO₂ anomalies in presence of polar stratospheric clouds, a case for which discussion is avoided in the paper, seems not to reflect the denitrification expected in this case.*

The reasons for the variations in AOD at the Southern Hemisphere high latitudes are unknown. In order to remove profiles potentially affected by PSCs from the analysis, layers of extinction profiles that were measured at temperatures < 195 K, the temperature for PSC formation, were removed from the analysis. The high monthly mean AODs at these latitudes were not removed with this new filter. Since the reasons for these fluctuations are unknown, we have decided to remove 60°S to 90°S from the analysis. We have adjusted the figures accordingly and have added the following text to the first paragraph of Sect. 3.1

"Latitudes south of 50°S were excluded from the analysis because there is no evidence of volcanic aerosol at these southern hemisphere high latitudes and OSIRIS AODs are dominated by seasonal variation."

*Overall, the paper provides interesting observations, but I think that two aspects have not been considered appropriately and might explain the inconsistencies found between MIPAS and OSIRIS measurements.*

*B. First, the authors consider monthly zonal means, with 10° latitude bins. This is a very coarse grid to study volcanic plume such as the ones considered in this study. The effects the authors want to study are largely diluted in the bin averaging and the mean values obtained from the binning are likely to be biased depending on the coverage of the instrument. Obviously, the effect of the bias is expected to increase if the comparison concerns data from two different instrument (with different coverage). Another similar source of bias could be the use of climatological data for ozone, neglecting the effects of local chemistry*

The 10° monthly latitude bins were selected for two reasons:

1. Bins of this size are often used in studies of long-term variations and trends. Therefore, the results of this work is more directly applicable when understanding, e.g., long-term trends in $NO_2$ timeseries.

2. This allowed a large amount of data to be averaged in most bins (typically ~100 measurements for both OSIRIS and MIPAS), which helped to decrease noise and smear out short-term natural variation. This is particularly useful when fitting contributions from QBO and estimating baseline $NO_2$, which varies seasonally and with latitude.

In order to test the effect of the bin sizes on our ability to assess processes around volcanic aerosol, we recalculated OSIRIS AODs in 5° latitude bins at a sampling of four times per month (approximately weekly). Based on the OSIRIS sampling, these are the smallest bins that would yield reasonable results. The observed AODs are shown in Figure 1 for latitude bins centered around 60°N, a latitude at which there were several episodes of enhanced volcanic aerosol. The maximum AODs for both bin sizes are very similar, suggesting that the smaller bin sizes do not give additional information on the volcanic events. Similar results were observed at other latitudes. The maximum observed AOD for all latitude/time bins was 0.0086 for the 10° monthly bins and was 0.0088 for the 5°, $\frac{1}{4}$ month latitude bins.

The 10° monthly bins are suitable for this analysis because the lifetime and transport of sulfate is slow compared with $SO_2$ and ash. Furthermore, individual OSIRIS profiles with very large extinctions ($>2\times10^{-3}$ km$^{-1}$), which could be associated with ash or very fresh plumes (~1-2 weeks) were removed from this analysis, which allowed for the direct focus on sulphate/$NO_2$ chemistry.

An example of the MIPAS sampling relative to OSIRIS AOD sampling is given in Figure 2 and Figure 3 for the 50°N latitude bin. This latitude was chosen to explore sampling because this latitude had valid MIPAS measurements for larger OSIRIS AODs (see Fig. 4a of the paper). Both OSIRIS AODs and MIPAS $NO_2$ measurements are sampled fairly consistently in latitude and time, with no obvious biases that would skew

the relationship between MIPAS $NO_2$ anomalies and OSIRIS partial column AODs, nor that would lead to large biases in ozone between the MIPAS and OSIRIS sampling. Furthermore, comprehensive tests on MIPAS and OSIRIS sampling have been performed for ozone by Toohey et al. (J. Geophys. Res., 118 (11), pp847–11,862, doi:10.1002/jgrd.50874, 2013) and found that MIPAS and OSIRIS have sampling biases of < 2% and < 5% for most latitudes/altitudes.

We have added the following text to the paper to address these comments:

- Sect. 3.1, second paragraph "Bins were tested for smaller latitude and time ranges, but yielded similar ranges of AODs, suggesting that smaller bin sizes did not capture more detailed processes in the volcanic plume."

- Sect. 4.1, fifth paragraph "These discrepancies could not be attributed to differences in sampling between OSIRIS and MIPAS, since MIPAS and OSIRIS both sample throughout the monthly 10° latitude bins. MIPAS measurements are not clustered in parts of the bin where smaller OSIRIS AODs were observed."

*C. The second weakness of the paper, in my opinion, concerns the discussion of the comparison between OSIRIS and MIPAS data, where arguments based on the limited degrees of freedom provided by MIPAS are used to justify the disagreement between results obtained from both instruments. Before drawing definitive conclusions from this hypothesis, the authors should check their interpretation by degrading both datasets to similar resolutions. These issues should be addressed before publication of the manuscript.*

In order to test the effect of the MIPAS resolution on the data, representative MIPAS $NO_2$ averaging kernels, A, were considered for the 50°N and 0° latitude bins and applied to the OSIRIS $NO_2$ profiles. The representative profiles had averaging kernel diagonal elements ($AKD$) closest to the mean averaging kernel diagonal elements ($\overline{AKD}$) within the vertical range of interest that the VCDs were calculated over. E.g., the representative MIPAS averaging kernels minimized

$$\sum_j \left( AKD(z_j) - \overline{AKD}(z_j) \right)^2 ,$$

where $z_j$ are altitude layers in the 5-km altitude range used to calculate the VCDs. The representative averaging kernels, in Figure 4, show reduced resolution over the altitude ranges used to calculate the partial columns in this paper, indicated by the dashed lines.

Since MIPAS $NO_2$ retrievals are logarithmic, smoothed OSIRIS $NO_2$ profiles, $x_s$, could be calculated using from original OSIRIS profiles, $x$, using

$$x_s = e^{A \cdot ln(x) + (1-A) \cdot ln(x_a)} ,$$

where $x_a$ is the MIPAS a priori. It then follows that the OSIRIS percent difference profiles, PD, can be interpolated to the MIPAS grid smoothed using

$$PD_s = e^{A \cdot ln(PD+1)} - 1 .$$

Figure 5 and Figure 6 show the unsmoothed and smoothed OSIRIS profiles for $0°$ latitude and $50°$N latitude, respectively. The dashed lines indicate the altitudes over which the partial columns were calculated. The unsmoothed OSIRIS profiles are what is shown in Fig 6 and Fig 7 of the paper respectively, but are presented here as a line plot so that the values of the $NO_2$ percent differences can be compared more closely. For both latitude ranges, the magnitude of negative $NO_2$ percent differences is reduced after applying the MIPAS averaging kernels. For example, at $50°$N, $NO_2$ percent differences of approximately -45% to -50% are smoothed to approximately -30% after the MIPAS averaging kernel is applied. Therefore these tests demonstrate that the resolution of MIPAS decreases the observed $NO_2$ anomalies under the presence of volcanic aerosol.

These tests could not be applied quantitatively to the entire analysis because the MIPAS averaging kernels vary for each profile and are correlated with the amount of $NO_2$ present. Individual MIPAS averaging kernels could not be applied directly to the OSIRIS data because MIPAS averaging kernels are only available for the measure-

ment times and locations of MIPAS, which are not coincident with OSIRIS measurement times and locations. This is why representative averaging kernels were used to assess this problem. The mean DOFS of the MIPAS VCDs tended to be lower during periods with enhanced OSIRIS partial column AODs, described in the text of the paper and shown in Figure 7. This decreased resolution would further smooth out the MIPAS measured $NO_2$ anomalies.

The following text has been added to Sect. 4.1, 5th paragraph:

"In order to test this, representative MIPAS averaging kernels were applied to the OSIRIS $NO_2$ percent difference profiles at 50°N and 0° latitudes. Representative averaging kernels were used because MIPAS $NO_2$ is retrieved in the logarithmic domain and the averaging kernel thus refers to the logarithm of the mixing ratio. By applying the averaging kernel directly to the percent difference profile, the MIPAS a priori profile does not need to be included in the calculations. The magnitude of the largest percent differences in the $NO_2$ percent difference profiles decreased from approximately -45% in the original OSIRIS profiles to approximately -30% in the smoothed OSIRIS profiles, demonstrating this damping effect. These tests did not account for variation of MIPAS DOFS with partial column AOD. The smaller DOFS observed for larger OSIRIS partial column AOD would lead to further damping of the MIPAS $NO_2$ percent differences. "

**Specific comments:**

*Abstract*

*L. 23: The authors mention percent difference of up to ~25%. They should mention with respect to what (to OSIRIS? To quiescent periods?)*

We have clarified this by adding "relative to baseline levels" to the text.

*1. Introduction*

*L. 8, p.1: The authors should define NOx like they defined NOy. Possibly at the same place to lighten the text, see technical comment on L. 4.*

A definition of NOx has been added

*L. 16, p.2: For the sake of clarity: "NOx/NOy decreased with increasing aerosol surface area".*

This has been edited as recommended

*2. Satellite and model datasets*

*2.3 Photochemical modelling*

All three reviewers had comments regarding the inputs to the photochemical model. In order to address these comments, we have changed the input ozone climatology and have revised our model errorbars using a series of perturbation tests. These tests are described in Sect. 2.3 and summarized in Table 1, and included in Fig. 5 of the revised paper. The break-down of the major contributions to the uncertainty estimates is given in Figure 8. These tests are described in Sect. 2.3 and summarized in Table 1, and included in Fig. 5 of the revised paper. The break-down of the major contributions to the uncertainty estimates is given in Figure 8. Aerosol (black) are for the two error terms related to extinction to surface area conversions. Ozone (blue), temperature (red), albedo (cyan), and NOy (green) refer are for year-to-year variability in these inputs. Systematic errors in ozone, temperature, albedo, NOy, $NO_2$ cross-section, and NO-$O_3$ reaction rates are mostly 1-2

*L. 16, p.4: Which type of climatological data are used for ozone and temperature? And why don't they use MIPAS ozone data which would be fully consistent with the used data for $SO_2$, $N_2O_5$, $HNO_3$ and $NO_2$ ? Perliski et al. (1989), cited in Randeniya et al. (op. cit., 1997), indicate that the ozone behaviour at high latitudes is expected to be dominated by local chemistry during summer, which is the time and region of interest for the present study, and where Eq. (2) is expected to have the greatest impact on ozone. Hence, using climatological data for the ozone field in this specific study focussing on the effect of volcanic eruptions might bias the results of the analysis.*

We initially used a standard $O_3$ climatology (from McPeters et al., JGR, 1999). However, we have since switched to an OSIRIS zonal, monthly-mean climatology for the revised version. (From previous experience we knew OSIRIS agreed very well with other ozone climatologies and so we simply used our model default inputs.) For temperature we use a climatology from Nagatani and Rosenfield (1993). OSIRIS does not measure temperature. We could have used MIPAS temperatures, and it has a small (2 K) bias in the lower stratosphere (Stiller et al., Atmos. Meas. Tech., 5, 289–320, 2012), but as demonstrated in the sensitivity study, systematic errors in temperature, make very little difference. The uncertainties associated with interannual variability in ozone were estimated and included in the errorbars in Fig. 5 of the revised paper. The contributions to these error estimates from ozone are shown in Figure 8, and are ~3-10% for most of the data points, except in the tropics (see Figure 8 of this reviewer response).

Nagatani, R. M., and J. E. Rosenfield (1993), Temperature, net heating and circulation, in The Atmospheric Effects of Stratospheric Aircraft: Report of the 1992 Models and Measurements Workshop, NASA Ref. Publ. 1292, edited by M. J. Prather and E. E. Remsberg, pp. A1–A47, NASA, Washington, D. C.

*L. 18, p.4: I don't understand what the authors mean with "but fixed to a specified Julian day". Please clarify.*

The model is integrated forward in time but instead of moving from JD to JD+1, where there would be differences in solar illumination, it again repeats JD. It continues to loop over this same day, up to 30 times, until the model converges. That is, concentrations at the end of the day match the beginning ("in a 24-hour steady-state").

The text has been modified to read "All remaining species are calculated to be in a 24-hour steady-state by integrating the model for as many as 30 days, but where the model remains fixed on the original, specified Julian day".

*L. 20, p.4: Since Thomason's climatology covers the whole period 1979-1995, it would*

*be useful to explain in a few words which data are used in the present study (specific year and/or region?). If they only use the approximated expressions (5) of Thomason's work as discussed in the following of the section, the authors could already announce that here (e.g.: "(...) using the aerosol surface area climatology of Thomason et al (1997) as explained later, (...)"). Actually, I don't know why the authors want to consider heterogeneous chemistry on background stratospheric aerosols, since they aim at studying typically volcanic situations. Besides, they mention further that they match the extinction coefficient with OSIRIS values, which should normally reflect this volcanic feature. All these points should be clarified.*

The sentence in question is an artifact of a much earlier draft of this paper that was not properly adjusted to reflect the methodology used herein. We did not use the Thomason et al. climatology at all, as the reviewer questions. Further, the use of the word "background" is also not correct in this context, also caught by the reviewer. The sentence now reads "Heterogeneous chemistry on stratospheric sulfate aerosols is included, but polar stratospheric clouds are not included."

*L. 24, p.4 to L. 10, p.5: The estimation of the SA in function of the extinction seems particularly crude, with a succession of approximations with choices which are not always clear nor convincing. The authors use the fact that in the case of Kasatochi, the mean particle size (see also next comment) decreases and that in the case of Sarychev, it increases, to choose a dependence between SA and the extinction which reflect none of these cases. This is a questionable way to approach this investigation focussing on a selection of recent volcanic eruptions (including those two ones). It is also worth to mention that Thomason et al (1997, op. cit.) uses such linear expression to describe cases where the extinction is higher than $2 . 10^{-2}$ km$^{-1}$, which is a really high value rarely encountered in the period considered here. The wavelength used to characterize the extinction in the equation in L. 26, p.4 is not mentioned, making it meaningless and preventing to compare this value with Thomason's coefficient (equal to 2000 to characterize the extinction at 1020 nm). They also "test" a non-linear SA*

*dependence mentioning Thomason's work, but their choices are quite different form what Thomason proposes. The choice of p=0.7 is quite similar to Thomason's one in the case of the weakest extinction values, but the choice p=1.3 is much higher than the value of 1 used with the highest extinction values. The final scaling "to account for potential errors" gets rid for good of any reference to real cases, making the reader definitively lost in the accumulation of assumptions.*

While it is true that in the case of Kasatochi and Sarychev one can attempt to model them in more detail based on the size distribution findings from the cases studies referenced (Sioris et al. and O'Neill et al.), this additional level of information is not available for most of the eruptions considered in this paper. But even this would be limited since the size distributions constantly evolved following the eruption (e.g., Kasatochi shifted the distribution towards smaller particles, and then later on, towards larger particles). Our goal was to use a consistent approach for all eruptions, and hence our "middle of the road" decision to use a linear k→SA relationship. While some of our choices for the sensitivity study are debatable, we opted to err on the conservative side (e.g., to consider a p=0.7 value which is more extreme than in Thomason et al.). This reflects the fact that not all values in the literature are directly applicable (as pointed out above) to our case, and that even these larger perturbations did not have a serious impact our modeled change-in-$NO_2$ – AOD relationships.

*L. 5, p.5: Sioris et al. (2010) don't claim that the stratospheric particle size decreases in the case of Kasatochi. They consider statistical values (through median radius and Angström exponent) which show a decrease of the averaged particle size. This is not the same thing. A particle size decrease supposes the occurrence of some evaporation process, while what happens here is the addition of a significant amount of very thin particles. The authors should change their formulation to avoid the confusion.*

We agree with the reviewer, and were simply sloppy with our wording here. We have adjusted this to read "For example, two months after the Kasatochi eruption, there was a shift in the ambient size distribution toward smaller particles (Sioris et al., 2010)

whereas Sarychev led to a shift towards larger particles (O'Neill et al., 2012)".

*L. 1, p. 5: The reference by Hansen and Travis, 1974 is a very interesting general reference on light scattering, but I don't see anything in this reference able to clarify the choices and formulation of the equation in L. 26, p.4. Hence, I am not sure it is really useful here.*

We assume you are referring to equation (3), on the bottom of page 4. The Hansen and Travis reference does not have this equation literally, although all the 'pieces' are there and it would have been left up to the reader to pull our equation out of this. In light of this, we explicitly derive our equation (3) in a short Appendix that was added to the manuscript and we reference it instead of Hansen and Travis.

*3. Calculation of monthly averages, anomalies, and baseline levels*

*3.1 OSIRIS and MIPAS*

*L. 13-20, p. 5: The use of monthly zonal means over latitude intervals as large as 10° might really bias the data and limit the quality of the correlative study of quantities derived from two different experiments such as OSIRIS and MIPAS in the present case. If as few as 5 measurements are possible for one bin, it could be possible, for instance, that one instrument covers only regions with low aerosol background while the other one catches the plume of an eruption. Or that one instrument covers a large region of the bin and returns average values of a quantity while the other one only catches some very local spot with specific (low or high) volcanic load. Did the authors prevent this kind of situation in some way?*

See response to General Comment B at the beginning of this reviewer response.

*L. 1 p. 7: Concerning the NO$_2$ response to the QBO, do the authors mean the fitted response as illustrated by the cyan curve in Figure 1b?*

Yes, we mean this and have replaced "the NO$_2$ response to the QBO" with "the fitted response to the QBO, as illustrated by the red dashed line in Fig. 1b" in the text to

clarify this.

*L. 11-15, p. 7: same remark as for L. 13-20, p. 5: I suppose that the model covers the whole 10° latitude monthly bin, while corresponding measurements might cover a reduced region of the bin, introducing potentially a bias in the analysis.*

The sampling of the OSIRIS measurements is fairly uniform across the bins, as demonstrated in the response to General Comment B, shown above in this document. Furthermore, the model data is calculated for a range of partial column AODs and then interpolated to the mean partial column AOD, as measured by OSIRIS. Therefore, the modelled $NO_2$ anomalies are for the mean partial column AODs as sampled by OSIRIS.

*3.3 Conversions between partial column AODs, aerosol extinction, and extinction at various wavelengths*

*L. 21-24, p. 7 (or L. 15-18, p. 5): I guess that profiles for which valid data don't cover the whole interval 3-7 km above the tropopause are rejected. This could be mentioned for the sake of completeness.*

We have clarified this in Sect. 3.1 with the following: "If a profile does not have valid data over all five measurement layers, it is not included in the analysis."

*4. Results*

*4.1 $NO_2$, $N_2O_5$, and $HNO_3$ VCDs*

*L. 26, p.8 and Figure 3: Overall, there is indeed a very clear correlation between the AOD enhancement and the $NO_2$ anomalies found by OSIRIS in the Northern latitudes. On the contrary, in the Southern polar latitudes, most of the time, no significant $NO_2$ decrease is found by OSIRIS and even more, strong local enhancements are observed each year. In the case of the Sounthern polar latitudes, the high AOD is most probably due to PSC, for which one would expect a denitrification, thus a decrease in $NO_2$. MIPAS $NO_2$ seems rather to behave in the opposite way, although the very limited coverage of the "volcanic regions" defined by the cyan curves would impose to be*

*cautious in any conclusion.*

See response to General Comment A, above.

*L. 2, p.9: I don't understand the reason given by the authors why the AOD should be excluded from the correlation before of its large variability. Large volcanic eruptions are able to produce an even larger variability.*

This text was not written very clearly. Data were not removed due to large variability in AODs, but because the large variability in AOD was not related to stratospheric aerosol. We have removed this statement from the text, as have removed data for 60°S to 80°S throughout the analysis, as described in the response to General Comment A, above.

*L. 6-7, p.9: Again, the authors consider monthly zonal bins with 10° latitude intervals for their analysis. This might be the reason for the apparently inconsistant behaviour (MIPAS vs OSIRIS, Northern latitudes vs. Southern latitudes) shown in Figure 3 (see comment on L.26, p.8 and Figure 3). Although the use of monthly zonal means is widely used in the community, as mentioned above, such large intervals are not well suited to study atmospheric processes at the level of a volcanic plume for such kind of eruptions, because the spatial extend and the temporal duration of the volcanic perturbation is relatively limited with respect to the spread of the bins. Hence, biases are potentially important and make any conclusion uncertain, especially if different instruments with different coverage and data rates are compared. The authors should remake their analysis by considering much shorter time and latitude intervals to see if the inconsistencies persist.*

See response to General Comment B, above.

*L. 9-15, p.9: The validity of the explanation given by the authors could be easily checked by degrading the OSIRIS using the MIPAS' averaging kernels (and possibly vice-versa). This way would allow comparing like with like. Did the authors make such check?*

[Figure]

See response to General Comment C, above.

*L. 29, p.9-L. 1, p. 10: The concept "somewhat linear" is strange. Overall, the interpretation of the shape of the curve is highly subjective, and the superposition of the modelled values on the plots biases our perception. As an example, the OSIRIS plots at 20°N show a behaviour which is neither linear at low AOD, nor saturated at the highest AOD values. The authors should remove these dubious interpretations.*

We have deleted the following "For smaller AODs (< ~3x10$^3$), this relationship appears to be somewhat linear. For larger AODs, however, this relationship displays greater curvature with additional aerosol having an increasingly smaller impact, which is consistent with the saturation at higher aerosol levels (e.g., Fahey et al., 1993)."

*L. 4-5, p.10: In the same way, the agreement in shape and in quantity between modelled and observed data is very relative, and in some case, quite bad. Hence, the authors should qualify their affirmation.*

We have replaced this sentence "The modelled data agrees well quantitatively with the OSIRIS measurements across latitudes and AOD ranges and reproduces the shape of the curve approaching saturation." with the following text:

"The modelled data agree well with the OSIRIS measurements and are within the estimated model errors for most OSIRIS data points."

We have also changed the abstract and conclusion to reflect this.

*4.2 OSIRIS NO$_2$ profiles*

*L. 25, p.10: The authors should remove the sentence "At this latitude, decreases in NO$_2$ are observed between ~10-20 km". This sentence is confusing, since in some cases (e.g. in 2002), an increase is observed in NO$_2$ instead of a decrease, and anyway, the next sentence expresses appropriately and in more detail what the authors mean.*

This has been deleted

*5. Conclusions*

*L. 20-21, P. 11: I think the conclusion concerning the influence of the DOFS on the disagreement between MIPAS and OSIRIS is premature, and should be verified as proposed above, before it is claimed.*

This has now been confirmed – see response to General Comment C.

*L. 22, p. 11: This line, with the qualification of "somewhat linear", should definitely be removed. The characterization of this relationship using the correlation coefficient is more than sufficient. In the same way, the expression "perfect linearity" on the next line should also be removed. As long as observations are concerned, there cannot be any perfect linearity.*

We have replaced "relationships. . . are somewhat linear" with "relationships. . . are observed".

We have removed the reference to "perfect linearity" and have replaced this with the following: "Heterogeneous chemistry becomes saturated toward larger aerosol concentrations (e.g.,Fahey et al., 1993) and can vary throughout the timeseries with other factors, such as temperature and available sunlight (e.g., Coffey, 1996), all of which can affect the linearity of the correlation."

*L. 27-28, p.11: The last sentence should be qualified according to the comment in L. 4-5, p.10.*

We have change this to "The modelled data agree well with the OSIRIS measurements and are within the estimated model errors for most OSIRIS data points."

**Technical corrections:**

*L.19, p.1: The authors could consider using "relationship" in the singular, or more precise, the word "correlation"?*

This has been changed

*L.23, p1: "periods affected by volcanic aerosol"*

This has been corrected

*L. 4, p.2: reformat the parenthese after BrONO$_2$ (no subscript). Writing "NOy species (where NOy = NOx +HNO$_3$+etc.+BrONO$_2$, and NOx=etc.; e.g. Coffey 1996)" might be more fluent. See also specific comment on L. 9.*

This has been changed as recommended

*L. 13, p.4: I suggest to write 10:00 LT to be consistent with the previous mention of time, and for the sake of clarity.*

This has been changed

*L. 6, p.5: "to keep the scattering efficiency constant".*

This has been corrected

*L. 15, p.8: It seems there is a problem of cross reference for Table 2.*

This has been corrected

*L. 13, p.10: "time [blanco] series"*

"timeseries" has been replaced with "time series" throughout
* * *
[Figure]

**Partial column AOD at 60 °N**

Fig. 1. OSIRIS partial column AOD at 60°N for the 10° monthly bins used in this paper and for smaller bins of 5° and $\frac{1}{4}$ of a month.

[Figure]

**Fig. 2.** OSIRIS partial column AOD (color-scale) and MIPAS NO2 measurement locations (black dots) versus time and latitude for data included in the 50°N latitude bin.

[Figure]

**Fig. 3.** As for Figure 2, for 2009 only.

[Figure]

[Figure]

[Figure]

**Fig. 4.** Columns of representative MIPAS averaging kernels at (top) 50°N and (bottom) 0°
latitude.

[Figure]

[Figure]

**Fig. 5.** Monthly mean NO2 percent difference profiles measured by OSIRIS (a) at original resolution and (b) smoothed using the OSIRIS averaging kernel at 0° latitude.

[Figure]

[Figure]

**Fig. 6.** As for Figure 5, for 50°N.

[Figure]

**Fig. 7.** For all latitude bins and dates included in the analysis, the mean MIPAS DOFS for NO2 versus the OSIRIS partial column AOD.

[Figure]

**Fig. 8.** Breakdown of parameters used to estimated total model uncertainty in Fig. 5 of the paper.

---

## Author Comment (AC2) · 11 Apr 2017

**Response to Referee 2**

We thank you for your comments, which have helped to improve our manuscript. Below we address the recommended changes point-by-point.

**1 General Comments**

*A big problem with the paper is that it totally relies on monthly zonal means which is not appropriate for the rather local and short lived volcanic plumes. Because of this, MIPAS SO$_2$ data are provided as 5-day means (Höpfner et al., 2015). Regarding the monthly zonal mean binning of AOD, smaller bins have been tested and found to have minimal*

*impact for the OSIRIS measurements. Please see response to General Comment B in response to Reviewer 3 for further details. We have elected to remove MIPAS SO$_2$ from the analysis, as described in more detail in responses below. Several important volcanic events listed there are missing or placed at the wrong latitude (Figure 3).*

Volcanic events are missing because we have only listed events that were observed in the OSIRIS stratospheric aerosol measurements. We have clarified this in the figure caption:

"The yellow triangles indicate the volcanic eruptions that were followed by significant increases in OSIRIS aerosol extinction, as listed in Table 2."

We have also clarified how volcanic eruptions are identified in other parts of the text, as noted in the details of this reviewer response below.

Thank you for noticing the problem with the latitudes of the volcanic eruptions in Fig. 3. The latitude of Sarychev Peak has been corrected in the revised figure.

*There is also no need to use inconsistent climatological ozone for the photochemistry, from both instruments, OSIRIS and MIPAS, are selfconsistent data available.*

Uncertainties due to the use of these climatologies have been estimated and are described in the text (Sect. 2.3), as well as in Figure 8 of response to Reviewer 3.

**2 Specific comments**

*Abstract: Do the authors mean a 4 km thick layer above the tropopause, in tropics and midlatitudes, and against what conditions is the change?*

We have re-worded this part of the abstract as follows:

"OSIRIS profile measurements indicate that the strongest correlations between NO$_2$ and volcanic aerosol extinction were for the 5-km layer starting 3 km above the mean tropopause at the given latitude. OSIRIS stratospheric NO$_2$ partial columns in this layer were found to be smaller than baseline NO$_2$ levels during these aerosol enhancements

by up to 60

*Please define NOx in line 9 of page 2.*

We have added the definition.

*Please improve wording in line 13 of page 3, it contains contradictions.*

We have reworded this to

"OSIRIS measures limb-scattered radiances from 82°S to 82°N, with nearly full coverage in the summer hemisphere."

*From the data version number, it looks like that Höpfner et al (2015) is used (line 9ff,page 4), here also the SO$_2$data prior to 2005 are OK. In these data, especially if the 5-day means are used, all important volcanic events should be identified with significance (see also lines 15 and 25, page 8). This is not the case for the older dataset presented in Höpfner et al (2013) which had the focus on the middle and upper stratosphere.*

MIPAS SO$_2$ played a very minor role in this paper. We included it because it was interesting to see how the SO$_2$ VCDs varied in the context of the OSIRIS AOD timeseries. Since we are using monthly averages for the other species in this study, we presented SO$_2$ in a consistent way. However, we don't want misrepresent the resolution/capabilities of MIPAS SO$_2$. Therefore, we have elected to remove MIPAS SO$_2$ from the paper. This means that panel b of Fig. 3 has been removed and that we have deleted the paragraph describing the MIPAS SO$_2$ and replaced it with a reference to Hopfner et al. 2015. This has not affected any of the major conclusions/results of this paper.

*For the crude assumptions on the Mie scattering efficiency the wavelength should be repeated (line 2, page 5). The statements on particle size (line 5, page 5) are confusing, more details please, give at least a range for the effective particle size. If you model particle size from aerosol formation from injected SO$_2$, you get in increase in effective*

*particle size for both volcanoes. What is the basis for the crude error assumptions (factor 3)?*

The basis for the large uncertainty we prescribe is two-fold: (1) We can only make an educated guess at the particles sizes and (2) since average particle sizes can increase or decrease following and eruption, depending on the relative amounts of new (smaller) particles formed vs. growth of existing particles. For a handful of eruptions there are some observations available, but most of those considered here this is not the case. This leads to uncertainty as to how to adjust the scattering efficiency. For (1) we conservatively estimated a factor of 3 based on sensitivity studies using different size parameters and distributions, and for (2) we estimated uncertainty based on Thomason et al. (1997). It is worth noting these uncertainties (presumably towards an upper limit) still do not have that large an impact on the model results, showing that the simulations are quite robust.

In response to the comment above we have fleshed this out with some additional detail/explanation.

"A scattering efficiency of 0.40 was calculated using Mie theory for background spherical sulfate particles (based on a log-normal distribution with size parameters of $r_g$=0.08 $\mu$m and $\sigma_g$=1.6). However, volcanic eruptions alter the size distribution, as $SO_2$ rapidly forms sulfuric acid, which can condense to form new small particles or increase the size of existing ones. This change in size distribution will affect the scattering efficiency, but the sign of this change is unknown. For example, two months after the Kasatochi eruption, there was a shift in the ambient size distribution toward smaller particles (Sioris et al., 2010) whereas Sarychev led to a shift toward larger particles (O'Neill et al., 2012)."

And

"To account for potential errors and variability over 2002-2015 in our background SA, we scaled 10000 by factors of 3 and 1/3. The large factor is based on the sensitivity of scattering efficiency to the aerosol size parameters for the particles sizes and

wavelengths considered here. For example, a change in effective radius by a factor of 2 leads to a change in scattering efficiency by a factor of 3 (see Hansen and Travis [1974] Figure 8)."

*Isn't there also an averaging kernel for OSIRIS (line 19ff, page 5)?*

There are not averaging kernels for OSIRIS retrievals. The resolution of OSIRIS is 2 km at the altitudes considered here, which is sufficient to retrieve partial column VCDs for the 5-km altitude range.

*In Fig. 1 an additional panel with the zonal wind at 20 km (?) might be useful (line 21, page 6).*

We have not included the zonal wind at 20 km since the QBO principal components are derived directly from the zonal winds.

*Why are different partial columns given in section 3.3 (line 23, page 7) and earlier in the text (including abstract)?*

The same partial column ranges are given in all section. The partial columns are calculated for the 5 measurement layers between 3-7 km above the mean thermal tropopause (e.g., over a 5-km range). We have clarified this in the text in Sect. 3.1.

". . . a 5-km altitude range starting at 3 km above the mean NCEP thermal tropopause at each latitude"

*In Table 2 at least the eruption of Rabaul in Oct. 2006 is missing.*

There was not a significant increase in OSIRIS aerosol extinction following the Rabaul eruption and therefore it, and likely other volcanoes, are not included in this table. We have added the following to the table caption to make sure this is clear:

"Note that this table only include eruptions that were followed by significant increases in OSIRIS aerosol extinction. Therefore it does not include all volcanoes known to have affected the stratosphere during this time-period."

*There appear to be contradictions between Fig. 3 and 4.*

I cannot find these apparent contradictions. The data presented in these figures come from the same data vectors, etc. We have revised the colorbar of Fig. 3 in order to make it easier to compare.

Improve Fig. 3 concerning $SO_2$ with the Höpfner (2015) 5-day dataset. We have removed MIPAS $SO_2$ from the figure and the paper, as described above.

*Fig. 3: Place the symbols for the volcanoes at the correct latitude.*

The latitude for Sarychev Peak has been corrected.

It might be better to use volume mixing ratios at 19 km (or 3 km above the tropopause) instead of the partial columns to reduce data gaps.

In earlier iterations of this analysis, we did consider VMRs at fixed altitude levels above the tropopause, but elected to present the results as partial columns. We have added the following text to the first paragraph of Sect 3.1 in order to explain this:

"Partial columns were used, instead of, e.g., volume mixing ratios at a fixed altitude, because the largest observed aerosol extinction ratios related to volcanic aerosol were observed at different altitude layers for different latitudes and times. The partial column altitude range was selected to include most of these large extinction ratios. Furthermore, MIPAS $NO_2$ measurements have low resolution at the altitudes affected by volcanic aerosol and therefore are better presented as partial columns."

*The results are also sensitive to the treatment of negatives in the individual data.*

We did not remove negative values in the individual profiles before averaging because we did not want to create a high bias in the datasets when averaging values that are close to zero.

*I don't understand the statement on tropospheric water vapor (line 22, page 8). The current understanding is that for explosive eruptions $SO_2$ is directly injected into the*

*stratosphere, in the plume only water from the volcano might matter, but the satellite sees only what comes out of the plume.*

Since we have removed $SO_2$ from the paper, this has been deleted.

*Section 4.2: In Figs. 1, 6 and 7 appear often extinction ratios < 1. Please explain or correct, from definition this should not happen. Please adjust color bars to reasonable range,*

The extinction ratio is defined as (Aerosol Extinction / Raleigh Extinction), in which case if the aerosol extinction is greater than the Rayleigh extinction, the ratio will be greater than 1.

We are also unsure what the reviewer means by "reasonable range". Therefore, we have adapted the color bars on various figures using the recommendations of Reviewer 1.

*Don't use formulations like 'somewhat linear' (line 4, page 10; line 3, page 12).*

This has been corrected, as per response to Reviewer 3.

**3 Technical corrections**

*Line 20ff, page 1: Better 'anticorrelation' instead of 'relationship'.*

This has been changed

*Line 6, page 5: typo and bad wording.*

We have reworded this paragraph, with the text starting with "In order to remove the seasonal variation from the $NO_2$ time series, the $NO_2$ anomaly ($dNO_2$) was calculated for each bin of the monthly mean $NO_2$ VCDs..."

*Figure 3, caption line 4: Do you mean a 4 km thick layer 3 km above the tropopause? Please improve text.*

This has been changed to "5–km layer starting 3 km above the thermal tropopause"

*Truncate Figs. 6 and 7 at 12 km, the data below are not relyable. Say 'aerosol extinction ratio' also in captions. The black contours are superfluous. The colorbars should have the same steps as the colors in the figures (less is more!).*

We have changed "extinction" to "extinction ratio" in the captions and the figure color bars and contours have been changed.

*Fig.8: Caption: Say 'correlation coefficient' instead of 'R'. The colorbar should have the same steps as the colors in the figure.*

The caption and figure have been changed as recommended.

―――――――――――――――

---

## Author Comment (AC3) · 11 Apr 2017

**Response to Referee 1**

We thank you for your comments, which have helped to improve our manuscript. Below we address the recommended changes point-by-point.

*The manuscript presents interesting and novel results, but important issues should be addressed before publication. The main one, in my opinion, is that the analysis relies on averages calculated into monthly 10° latitude bins. Using such large intervals in time and space to study quite short-lived and local events such as volcanic plumes can lead to important biases in the results.*

[Figure]

See response to General Comment B in response to Referee 3.

**Specific comments:**

*p.2, l.9: "NOx" should be defined.*

We have added a definition for NOx.

*p.4, l.18: Please give more details on the climatological profiles used for ozone and temperature, and explain your choice of using these profiles rather than using directly MIPAS and OSIRIS measurements of $O_3$ and T.*

This has been addressed in response to Reviewer 3 under Sect. 2.3.

*p.4, 19-20: Please clarify the sentence "All remaining species are calculated to be in a 24-hour steady-state by integrating the model over 30 days, but fixed to a specified Julian day".*

This has been addressed in response to Reviewer 3.

*p.4, l.22: The effects of polar stratospheric clouds are not considered here. However, these could play a role in the altitude range under consideration, especially in summer, which is the season on which this study is focused. This should be discussed while interpreting the results.*

The lack of PSCs should not be a large factor here since only latitudes north of 55S are considered in the updated analysis (see Response to Reviewer 3, General Comment A). Moreover, the intent of the model is to demonstrate that the 'average' relationship seen in the OSIRIS $NO_2$ and aerosol extinction observations can be captured by the model, thereby demonstrating that the two are consistent with our current understanding of the impact of aerosol on NOy partitioning. It is possible that one or more OSIRIS monthly/zonal means might be contain some PSC signal, and this would lead to some departure from the non-PSC expected relationship, but there is no evidence of this in the observations at 70° or 80°N and at the high end of the AOD scale (the last two

panels in Figure 5).

*p.5, l.9-10: How the factors 3 and 1/3, used to account for potential errors in your background aerosol surface area, have been chosen?*

The relationship between surface area and extinction, for Mie theory, is through the scattering efficiency, which depends on several factors. The most important of is the aerosol size distribution. As there is some uncertainty in the parameters than define this, such as the effective radius and the variance of the distribution, we considered a range of different values. The end results was the scattering efficiency was found to vary by a factor of 2 (1/2), and so we used a factor of 3 (1/3) to be conservative in our uncertainty estimates

*p.5, l.21: Place clarify "for the five measurement layers from 3-7 km above the NCEP thermal tropopause..." I do not clearly understand which measurement layers you are using.*

We are adding the partial column using the layers 3, 4, 5, 6, and 7 km above the tropopause. We have clarified this throughout the text by instead referring to the partial column over "the 5-km layer starting 3 km above the mean tropopause at the given latitude"

*Section 3: The analysis is based on monthly means calculated in $10°$ latitude bins. The choice of such large intervals in time and space does not seem to be the most appropriate to study volcanic plumes, which are quite short-lived and local events. A different distribution of the observations from OSIRIS and MIPAS in a given bin might lead to very different results. The same comment is also relevant for the photochemical model, which covers the whole bin in a uniform way, while this is probably not the case for the observations. Was it not possible to perform the analysis using more appropriate bins? In this case, the authors should find a way to estimate the sensitivity of their results to this problem, and this should be thoroughly discussed in the paper. Have the authors try to use only the modelled data in collocation in time and space with the*

*observations?*

For OSIRIS and the photochemical model, there should not be great sensitivity to this. OSIRIS $NO_2$ and extinction measurements are sampled approximately the same. For the photochemical model, we are interpolating $NO_2$ to the mean extinction as measured by OSIRIS.

MIPAS sampling is not identical to OSIRIS, but covers the selected bins fairly evenly. This has been discussed in Response to Reviewer 3, General Comment B.

*p.8, l.13-17 and Fig.3: Please comment the high AOD levels at southern high latitudes, which are obviously not due to volcanic eruptions. We can read, p.9 l. 5-6, that these are "perhaps due to polar stratospheric clouds". This should be discussed already in the description of Fig.3, and this statement should be explained, or at least associated with a reference.*

We have removed southern hemisphere high latitudes from analysis, as discussed in General Comment A in response to Referee 3.

*p.9, l8: Is the second interval considered to calculate the correlation coefficients 40°N-80°N or 40°S-80°N? The information given in the figure is inconsistent with what is said in the text. Same remark p.9, l.22.*

Thank you for noticing these typos. We have corrected the text to 40°N-80°N.

*p.9, l.16-18: This could be checked by applying MIPAS averaging kernels to the OSIRIS data. This has already been done in several studies comparing MIPAS data to other data sets characterised by a better vertical resolution.*

This has now been tested. See response to General Comment C in response to Referee 3.

*p.10, l.15-16: Was the vertical resolution of the MIPAS $N_2O_5$ profiles also not good enough to look at the effect of volcanic aerosol on this species as a function of altitude?*

The vertical resolution of $N_2O_5$ is 6-8 km for much of the lower stratosphere (see Fig. 3 of Mengistu Tsidu et al 2004) and therefore is not sufficient to look at the effect of volcanic aerosol. We have clarified this in the text in Sect. 4.2:

"The MIPAS $NO_2$ and $N_2O_5$ profiles were not considered because of limitations in the vertical resolution in the lower stratosphere."

**Technical corrections:**

*p.1, l.24: Remove "of"*

This has been removed

*p.1, l.25: Please change "relationship" to "anti-correlation"*

This has been replaced

*p.3, l.22, p.4, l.1, p.4, l.14 for example: Please write "lower than" or "greater than" instead of "<" or ">".*

We have replaced the ">" and "<" throughout the document.

*p.5, l.19: For the sake of clarity, please add "number densities" after "$SO_2$, $NO_2$, $N_2O_5$ and $HO_3$".*

This has been added

*p.6, l.13: "on the same order of magnitude AS the variation..."*

This has been corrected

*p.6, l.27: "...periods that WERE not affected by volcanic aerosol."*

This has been corrected

*p.10, l.17: Please add "percent difference" in "OSIRIS and modelled $NO_2$ percent dif-ference profiles".*

[Figure]

This has been corrected

*p.10, l.20-23: For the sake of clarity, the given percent different values should be negative (same remark p.11, l.2)*

We have specified all the percent differences as negative, as recommended

*Fig. 3, 6, 7 and 8: These figures would be clearer if there was a limited number of colours in the colour bar, so that it corresponds to the colour levels shown in the plots.*

We have changed the figures as recommended.

---

## Author Response (AR2)

**Response to Editor**

Thank you for noticing the typos listed below. We have corrected them all, as suggested.

*Technical corrections:*
*p2, l25: I guess it should read "in-situ" instead of "situ".*
*p3, l15: launched "on"?*
*p11, l25: rephrase sentence. Did you mean "which is why"?*
*p13, l2: Instead of just writing "R" I would suggest to write "correlation coefficient R"*

**Response to Referee #2**

We thank Referee #2 for the following comments and have addressed them as follows. The referee comments are given in italics.

*General Remarks*

*The manuscript has improved compared to the previous version. Instead of considering requests of the referees for additional remarks on effects of polar stratospheric clouds and on the coarse temporal and spatial resolution the parts sensitive to that were skipped. Because of this an additional sentence in the introduction like "This coarse resolution study is suitable for use with longterm chemistry climate model simulations or statistics but not for understanding processes" should be included.*

We have added the following text to the introduction to clarify

"This coarse resolution study cannot be used to understand short-term processes in the days immediately following a volcanic eruption. Instead, it can be used to understand the longer-term effect of volcanic aerosol in the months following an eruption."

*It would be useful to provide a similar analysis for the effects of temporal and spatial resolution for the tropics (0°) as the one for 60°N and polar day eruptions included in the interactive comment AC1 in an appendix or supplement.*

We have included an analysis for 0° in a supplement.

*Also, the original papers (Randel and Wu, 1996, and or Wallace et al., 1993, both J. Atmos. Sci.) on the QBO-fit should be cited for better understanding.*

We have added a reference to Randel and Wu.

*\section{Specific remarks}*

*Abstract and other sections: How is 'mean tropopause' defined in midlatitudes where the tropopause height is varying with the Rossby-waves? Do you mean 'climatological mean tropopause height' here, independent of the observed air mass?*

Yes, we mean the climatological mean tropopause height.  We have clarified this in the text.

*'Baseline NO_2' is mentioned very often in the text but is not defined in enough detail. In Section 3.1 some crude information is given but it would be better to provide explicitly the used time periods for better explanation of the odd positive anomalies in the figures. I would prefer to use 'background NO_2'. It might be useful to show it in a figure. Is it different for the instruments and the model?*

We have changed baseline to background, as recommended.  The background levels are calculated for the time-periods determined by the AOD and extinction ratio thresholds bins that are not affected by volcanic aerosol (e.g., time-periods varies by latitude).  We have attempted to clarify this throughout the text.

We have added the following text to Sect. 3.1:

"Background levels are the monthly mean values in the time series at a given latitude for months that were not affected by volcanic aerosol, based on a threshold for OSIRIS aerosol measurements."

As well as this further in Sect. 3.1

"For example, the background $NO_2$ VCD for January at 0° is the average of all January VCD measurements at 0°, except for the measurements taken in 2003, 2010, and 2012, when volcanic aerosol was enhanced.  The QBO signal was added to the background $NO_2$ to account for year-to-year variations in background $NO_2$ (e.g., January 2008 background is slightly different than January 2009 due to QBO)."

We have included a bit more information on the model in Sect. 3.2 to further explain the model results

"The background partial column AOD was calculated from OSIRIS measurements at the given latitude, for AODs that were less than $2 \times 10^{-3}$.  For the profiles, a similar interpolation procedure was used at each altitude layer using OSIRIS-measured aerosol extinction and the extinction ratio threshold of 1.2 to identify influence from volcanoes."

We have also added some text describing the positive anomalies in the model data in Sect. 4.2

"This leads to positive anomalies in $NO_2$ during periods with lower levels of volcanic aerosol, particularly before 2005.  These positive anomalies are also apparent in the photochemical model results, which vary based entirely on aerosol levels.  The photochemical model calculated background $NO_2$ is interpolated to the mean background aerosol as measured by OSIRIS over the full measurement period (e.g., the average aerosol extinction for January, when aerosol extinction ratios are below the threshold of 1.2).  Therefore, early in the time series, the modelled $NO_2$ is calculated for lower levels of aerosol

while background $NO_2$ is calculated for slightly higher levels of aerosol, leading to positive modelled $NO_2$ anomalies."

*In Eqn. 3 better write '4 $\cdot$ 1000' and mention the unit conversion factor 1000 here (can be skipped in appendix then).*

This has been changed as suggested.

*Table 3 and Figure 3: It is odd that the strongest eruption in the tropics in the MIPAS period, Rabaul sometimes called Tavurvur, 7 Oct 2006, 4S/152E, see e.g. Santer et al., 2014, Nature Geoscience), is not listed. In Fig. 3 it is close to the second strongest signal in AOD but also NO_2 (combined effect with Soufriere Hills?). Details and further references on that see Höpfner et al. (2015). It might be also worth to check if the eruption of Jebel el Tair (16N/42E, 30 Sep 2007, northward transport of aerosol as for Nabro) is significant.*

We agree that it could be a combined effect with Soufriere Hills and Rabaul. We have therefore added Rabaul to Table 3 and Figure 3 and the text.

We do not see any evidence of the Jebel el Tair volcano. This may be because OSIRIS does not measure in the winter hemisphere.

*Conclusions, paragraph 1: mention analysed latitudes!*

This has been added.

*\section{Technical corrections}*

*Page 2, line 14: Adjust text to the removal of analysed data
(vortex data were not analysed).*

This has been changed as recommended.

*Page 7, line 26: I suppose it should be 40°S -- 40°N.*

This has been corrected.

[revised manuscript text omitted]